# Robust and Data-efficient Q-learning by Composite Value-estimation

**Gabriel Kalweit**  *kalweitg@cs.uni-freiburg.de*
**Maria Kalweit**  *kalweitm@cs.uni-freiburg.de*
**Joschka Boedecker**  *jboedeck@cs.uni-freiburg.de*
*Neurorobotics Lab and BrainLinks-BrainTools*
*University of Freiburg*
*Germany*

**Reviewed on OpenReview:** *https://openreview.net/forum?id=ak6Bds2DcI*

## Abstract

In the past few years, off-policy reinforcement learning methods have shown promising results in their application to robot control. Q-learning based methods, however, still suffer from poor data-efficiency and are susceptible to stochasticity or noise in the immediate reward, which is limiting with regard to real-world applications. We alleviate this problem by proposing two novel off-policy Temporal-Difference formulations: (1) Truncated Q-functions which represent the return for the first $n$ steps of a target-policy rollout with respect to the full action-value and (2) Shifted Q-functions, acting as the farsighted return after this truncated rollout. This decomposition allows us to optimize both parts with their individual learning rates, achieving significant learning speedup and robustness to variance in the reward signal, leading to the Composite Q-learning algorithm. We show the efficacy of Composite Q-learning in the tabular case and furthermore employ Composite Q-learning within TD3. We compare Composite TD3 with TD3 and TD3($\Delta$), which we introduce as an off-policy variant of TD($\Delta$). Moreover, we show that Composite TD3 outperforms TD3 as well as TD3($\Delta$) significantly in terms of data-efficiency in multiple simulated robot tasks and that Composite Q-learning is robust to stochastic immediate rewards.

## 1 Introduction

In recent years, Q-learning (Watkins & Dayan, 1992) has achieved major successes in a broad range of areas by employing deep neural networks (Mnih et al., 2015; Silver et al., 2018; Lillicrap et al., 2016), including environments of high complexity (Riedmiller et al., 2018) and even in first real-world applications (Haarnoja et al., 2019). Deep Q-learning methods, however, are still far from being easily applicable to solving complex real-world tasks due to their high demand for data samples and their lack in learning stability partially caused by the stochastic nature of transitions in real-world settings, such as in robotics, intelligent machine-brain interfaces or health applications. Especially stochasticity in the immediate reward signal itself may cause policies to converge to suboptimal performance. This stochasticity can be induced by multiple sources of randomness, for example through stochastic next state transitions due to sensor or actuator noise, occlusions, or due to data-driven reward estimation (Mees et al., 2020; Ni et al., 2020; Fu et al., 2018). This increases the complexity of value-estimation tremendously due the long temporal horizon the reward signal has to propagate through, possibly depending on many decision steps and transitions. To overcome the complexity of growing task horizon, in this work, we propose to break down the long-term return into a composition of several short-term predictions over a *fixed temporal horizon*. We approach this via a new formulation of consecutive bootstrapping from value functions corresponding to different horizons of the target-policy associated to the *full return*. For its formulation, we define *Truncated Q-functions*, representing the return for the first $n$ steps of a target-policy rollout with respect to the full action-value. In

addition, we introduce *Shifted Q-functions* which represent the farsighted return after this truncated rollout. Both are then combined in a recursive definition of the Q-function for the final algorithm. Starting from a reformulation of the vanilla Q-learning formalism, our approach offers the flexibility to separately optimize hyperparameters for different parts of value-estimation, showing that Shifted Q-functions suffer less from variance compared to the full return thus allowing for higher learning rates. Most similar to our approach is TD($\Delta$) (Romoff et al., 2019), composing the full return of value functions with increasing discount. Increasing discount factors translate to growing task horizons, leading to a decomposition of the long-term prediction task into multiple subtasks of smaller complexity. In their work, Romoff et al. (2019) argue that values for smaller discount values can potentially stabilize more quickly, leading to faster and more stable learning. Since the values corresponding to larger discount only represent the residual to the values of smaller discount, however, TD($\Delta$) does not alleviate the issues arising with stochastic rewards. Other techniques for variance reduction in TD-learning are reward estimation (Romoff et al., 2018) or averaging value functions (Anschel et al., 2017). Both techniques are orthogonal to our work and can be combined with Composite Q-learning to possibly further enhance learning stability. However, as we show in this work, the increase in performance and data-efficiency comes from the possibility of higher learning rates in the long-term Shifted Q-functions and not the short-sighted Truncated Q-functions. It can thus be argued that reward estimation and averaging come at a cost of slower convergence. Asis et al. (2020) propose to learn action-values for a fixed-horizon via consecutive bootstrapping. In their formulation, however, the consecutive value functions are estimated over different target-policies with respect to the different horizons and ignore the residual of the full return. Thus, they can be non-optimal with respect to the task at hand.

Our contributions are fourfold. First, we introduce the *Composite Q-learning* algorithm. Second, we show that the targets of Shifted Q-functions suffer less from variance compared to the full action-value, thus allowing for higher learning rates. Third, we introduce *TD3($\Delta$)*, an off-policy extension of TD($\Delta$) to deep Q-learning. And fourth, we employ Composite Q-learning within TD3 and evaluate Composite TD3 in various simulated robot tasks under normal and noisy reward functions.

## 2    Background

We consider tasks modelled as Markov decision processes (MDP), where an agent executes action $a_t \in \mathcal{A}$ in some state $s_t \in \mathcal{S}$ following its stochastic policy $\pi : \mathcal{S} \times \mathcal{A} \to [0, 1]$. According to the dynamics model $\mathcal{M} : \mathcal{S} \times \mathcal{A} \times \mathcal{S} \to [0, 1]$ of the environment, the agent transitions into some state $s_{t+1} \in \mathcal{S}$ and receives scalar reward $r_t$. The agent aims at maximizing the expected long-term return $V^\pi(s_t) = \mathbf{E}_{a_{j \geq t} \sim \pi, s_{j > t} \sim \mathcal{M}}[\sum_{j=t}^{T-1} \gamma^{j-t} r_j | s_t]$, where $T$ is the (possibly infinite) temporal horizon of the MDP and $\gamma \in [0, 1]$ the discount factor. It therefore tries to find $\pi^*$, s.t. $V^{\pi^*} \geq V^\pi$ for all $\pi$. If the model of the environment is unknown, model-free methods based on the Bellman Optimality Equation over the so-called action-value $Q^\pi(s_t, a_t) = \mathbf{E}_{a_{j>t} \sim \pi, s_{j>t} \sim \mathcal{M}}[\sum_{j=t}^{T-1} \gamma^{j-t} r_j | s_t, a_t]$, can be used, $Q^*(s_t, a_t) = r_t + \gamma \max_a \mathbf{E}_{s_{t+1} \sim \mathcal{M}}[Q^*(s_{t+1}, a)]$. In the following, we abbreviate $\mathbf{E}_{a_{j>t} \sim \pi, s_{j>t} \sim \mathcal{M}}[\cdot | s_t, a_t]$ by $\mathbf{E}_{\pi, \mathcal{M}}[\cdot]$. One of the most popular model-free reinforcement learning methods is the sampling-based off-policy *Q-learning* algorithm (Watkins & Dayan, 1992). A representative of continuous model-free reinforcement learning with function approximation is the *Deep Deterministic Policy Gradient* (DDPG) actor-critic method (Lillicrap et al., 2016). In DDPG, actor $\mu$ is a deterministic mapping from states to actions, $\mu : \mathcal{S} \to \mathcal{A}$, representing the actions that maximize the critic $Q$, i.e. $\mu(s_t) = \arg\max_a Q(s_t, a)$. $Q$ and $\mu$ are estimated by function approximators $Q(\cdot, \cdot | \theta^Q)$ and $\mu(\cdot | \theta^\mu)$, parameterized by $\theta^Q$ and $\theta^\mu$. The critic is optimized on the mean squared error between predictions $Q(s_j, a_j | \theta^Q)$ and targets $y_j = r_j + \gamma Q'(s_{j+1}, \mu'(s_{j+1} | \theta^{\mu'}) | \theta^{Q'})$, where $Q'$ and $\mu'$ are target networks, parameterized by $\theta^{Q'}$ and $\theta^{\mu'}$. The parameters of $\mu$ are optimized following the deterministic policy gradient theorem (Silver et al., 2014), $\nabla_{\theta^\mu} \hookleftarrow \frac{1}{m} \sum_j \nabla_a Q(s, a | \theta^Q)|_{s=s_j, a=\mu(s_j | \theta^\mu)} \nabla_{\theta^\mu} \mu(s | \theta^\mu)$, and the parameters of the target networks are updated via Polyak averaging, i.e. $\theta^{Q'} \hookleftarrow (1-\tau)\theta^{Q'} + \tau\theta^Q$ and $\theta^{\mu'} \hookleftarrow (1-\tau)\theta^{\mu'} + \tau\theta^\mu$, with $\tau \in [0, 1]$. The state-of-the-art actor-critic method TD3 (Fujimoto et al., 2018) then adds three adjustments to vanilla DDPG. First, the minimum prediction of two distinct critics is taken for target calculation to alleviate overestimation bias, an approach belonging to the family of Double Q-learning algorithms (van Hasselt et al., 2016). Second, Gaussian smoothing is applied to the target-policy, addressing the variance in updates. Third, actor and target networks are updated every $d$-th gradient step of the critic, to account for the problem of moving targets.

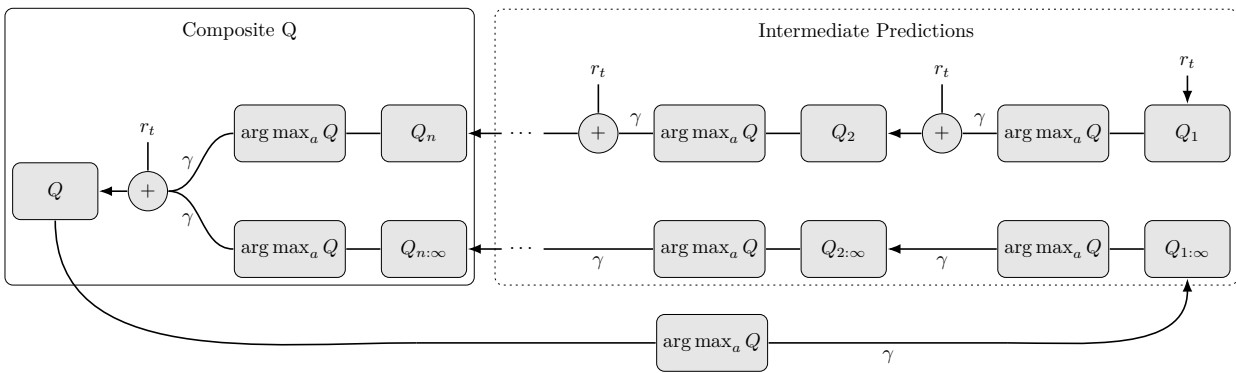

Figure 1: Structure of Composite Q-learning. $Q_i$ denote the Truncated and $Q_{i:\infty}$ the Shifted Q-functions at step $i$. $Q$ is the complete Composite Q-function. Directed incoming edges yield the targets for the corresponding value function. Edges denoted by $\gamma$ are discounted.

## 3 Combining Long-term and Short-term Predictions in Q-learning

In the following, we introduce the Composite Q-learning algorithm, along with Off-policy TD($\Delta$) as an additional baseline. Both approaches aim at decomposing the long-term value into values of smaller time scales. Composite Q-learning approaches this by dividing the bootstrapping into a short-term and a long-term component. Off-policy TD($\Delta$) on the other hand formalizes a delta function to estimate the remainder of an action-value corresponding to a smaller discount factor.

### 3.1 Composite Q-learning

We estimate the return of $n$-step rollouts of the target-policy via Truncated Q-functions which we then combine to the full return with model-free Shifted Q-functions, an approach we call *Composite Q-learning*, while remaining *purely off-policy*. Since these quantities cannot be estimated directly from single-step transitions, we introduce a consecutive bootstrapping scheme based on intermediate predictions. Put differently, Composite Q-learning *does not* learn from $n$-step returns, but rather learns a decomposition of the full Q-value on the basis of *single-step* transitions alone. For an overview, see Figure 1. Code based on the implementation of TD3[1] can be found in the supplementary[2].

### 3.1.1 Truncated Q-functions

In order to decompose the action-value into multiple truncated returns, assume that $n \ll (T-1)$ and that $(T-1-t) \mod n = 0$ for task horizon $T$. We make use of the following observation:

$$
\begin{aligned}
Q^\pi(s_t, a_t) &= \mathbf{E}_{\pi,\mathcal{M}} \left[ r_t + \gamma r_{t+1} + \gamma^2 r_{t+2} + \gamma^3 r_{t+3} + \cdots + \gamma^{T-1} r_{T-1} \right] \\
&= \mathbf{E}_{\pi,\mathcal{M}} \Bigg[ \left( \sum_{j=t}^{t+n-1} \gamma^{j-t} r_j \right) + \gamma^n \left( \sum_{j=t+n}^{t+2n-1} \gamma^{j-(t+n)} r_j \right) \\
&\qquad\qquad + \cdots + \gamma^{T-n} \left( \sum_{j=T-n}^{T-1} \gamma^{j-(T-n)} r_j \right) \Bigg].
\end{aligned}
\tag{1}
$$

That is, we can define the action-value as a combination of partial sums of length $n$. We can then define the Truncated Q-function as $Q_n^\pi(s_t, a_t) = \mathbf{E}_{\pi,\mathcal{M}}[\sum_{j=t}^{t+n-1} \gamma^{j-t} r_j]$, which we plug into Equation (1):

$$
Q^\pi(s_t, a_t) = \mathbf{E}_{\pi,\mathcal{M}}[Q_n^\pi(s_t, a_t) + \gamma^n Q_n^\pi(s_{t+n}, a_{t+n}) + \cdots + \gamma^{T-n} Q_n^\pi(s_{T-n}, a_{T-n})].
\tag{2}
$$

---

[1] https://github.com/sfujim/TD3
[2] https://github.com/NrLabFreiburg/composite-q-learning

**Proposition 1.** *Let $Q_1^\pi(s_t, a_t) = r_t$ be the one-step Truncated Q-function and $Q_{i>1}^\pi(s_t, a_t) = r_t + \gamma \mathbf{E}_{t,\pi,\mathcal{M}}[Q_{i-1}^\pi(s_{t+1}, a_{t+1})]$ the i-step Truncated Q-function. Then $Q_i^\pi(s_t, a_t)$ represents the truncated return $Q_i^\pi(s_t, a_t) = \mathbf{E}_{t,\pi,\mathcal{M}}[\sum_{j=t}^{t+i-1} \gamma^{j-t} r_j]$.*

See the proof of Proposition 1 in the appendix. Please note, that the assumption of $(T - 1 - t) \mod n = 0$ only leads to an easier notation and is not a general restriction. If $(T-1-t) \mod n \neq 0$, then the last partial sum simply has a shorter horizon. On the basis of Proposition 1, in the following we estimate $Q_n^*(s_t, a_t)$ off-policy via consecutive bootstrapping and stochastic approximation. In order to limit the prediction to horizon $n$, we estimate $n$ different truncated value functions belonging to increasing horizons, with the first one estimating the immediate reward function. All consecutive value functions then bootstrap from the prediction of the preceding value function evaluating the target-policy of *full* return $Q$ for a fixed horizon. Analogously to vanilla Q-learning, the update procedure is given by:

$$
\begin{aligned}
Q_1(s_t, a_t) &\leftarrow (1 - \alpha_{\text{Tr}})Q_1(s_t, a_t) + \alpha_{\text{Tr}} r_t \text{ and} \\
Q_{i>1}(s_t, a_t) &\leftarrow (1 - \alpha_{\text{Tr}})Q_i(s_t, a_t) + \alpha_{\text{Tr}}(r_t + \gamma Q_{i-1}(s_{t+1}, \arg\max_a Q(s_{t+1}, a))),
\end{aligned}
\tag{3}
$$

with learning rate $\alpha_{\text{Tr}}$. Please note that in order to estimate Equation (2), the dynamics model would be needed to get $s_{t+c\cdot n}$ of a rollout starting in $s_t$. In the following, we describe an approach to achieve an estimation of Equation (2) model-free.

### 3.1.2 Shifted Q-functions

To get an estimation for the remainder of the rollout $Q_{n:\infty}^\pi = \mathbf{E}_{\pi,\mathcal{M}}[\gamma^n Q(s_{t+n}, a_{t+n})]$ after $n$ steps, we use a consecutive bootstrapping formulation of the Q-prediction as a means to skip the first $n$ rewards of a target-policy rollout.

**Proposition 2.** *Let $Q_{1:\infty}^\pi(s_t, a_t) = \mathbf{E}_{t,\pi,\mathcal{M}}[\gamma Q^\pi(s_{t+1}, a_{t+1})]$ be the one-step Shifted Q-function and $Q_{i>1:\infty}^\pi(s_t, a_t) = \mathbf{E}_{t,\pi,\mathcal{M}}[\gamma Q_{i-1:\infty}^\pi(s_{t+1}, a_{t+1})]$ the i-step Shifted Q-function. Then $Q_{i:\infty}^\pi(s_t, a_t)$ represents the shifted return $Q_{i:\infty}^\pi(s_t, a_t) = \mathbf{E}_{t,\pi,\mathcal{M}}[\gamma^i Q^\pi(s_{t+i}, a_{t+i})]$.*

The proof of Proposition 2 can be found in the appendix. Analogously to the consecutive bootstrapping formulation of Truncated Q-functions, the updates for the Shifted Q-functions are:

$$
\begin{aligned}
Q_{1:\infty}(s_t, a_t) &\leftarrow (1 - \alpha_{\text{Sh}})Q_{1:\infty}(s_t, a_t) + \alpha_{\text{Sh}}(\gamma \max_a Q(s_{t+1}, a)) \text{ and} \\
Q_{(i>1):\infty}(s_t, a_t) &\leftarrow (1 - \alpha_{\text{Sh}})Q_{i:\infty}(s_t, a_t) + \alpha_{\text{Sh}}(\gamma Q_{(i-1):\infty}(s_{t+1}, \arg\max_a Q(s_{t+1}, a))),
\end{aligned}
\tag{4}
$$

with learning rate $\alpha_{\text{Sh}}$. The variance of the remainder of the rollout decreases with later initial point in time:

$$
\begin{aligned}
\text{var}[\gamma^i Q^\pi(s_{t+i}, a_{t+i})] &= \text{var}[\gamma^i(r_{t+i} + \gamma Q^\pi(s_{t+i+1}, a_{t+i+1}))] \\
&= \text{var}[\gamma^i r_{t+i}] + \text{var}[\gamma^{i+1} Q^\pi(s_{t+i+1}, a_{t+i+1})] \\
&\quad + 2\,\text{cov}[\gamma^i r_{t+i}, \gamma^{i+1} Q^\pi(s_{t+i+1}, a_{t+i+1})].
\end{aligned}
\tag{5}
$$

If $\text{cov}[\gamma^i r_{t+i}, \gamma^{i+1} Q^\pi(s_{t+i+1}, a_{t+i+1})] \geq 0$, it holds that:

$$
\text{var}[\gamma^i Q^\pi(s_{t+i}, a_{t+i})] \geq \text{var}[\gamma^{i+1} Q^\pi(s_{t+i+1}, a_{t+i+1})].
\tag{6}
$$

Please note that $\gamma^i Q^\pi(s_{t+i}, a_{t+i})$ corresponds to Shifted Q-functions at different stages $i$ of shifting, with $i = 0$ for the full Q-function. It hence follows:

$$
\text{var}[Q^\pi(s_t, a_t)] \geq \text{var}[Q_{1:\infty}^\pi(s_t, a_t)] \geq \text{var}[Q_{2:\infty}^\pi(s_t, a_t)] \geq \cdots \geq \text{var}[Q_{n:\infty}^\pi(s_t, a_t)].
\tag{7}
$$

The covariance of reward and future return is likely to be positive if the immediate reward depends on the next state (Romoff et al., 2018). Otherwise the covariance is zero. Hence, it can be argued that this inequality

is strict if the variance of the reward is larger than zero. This also holds for the case of zero variance for the immediate reward, if the immediate reward depends on a stochastic next state. We therefore argue that in these settings, as very common in real-world experiments, Shifted Q-functions allow for higher learning rates compared to the full Q-function, since this translates to lower variance in their gradients. We empirically validate this finding in our experiments in Section 4.

### 3.1.3 Composition

Following the definitions of Truncated and Shifted Q-functions, we can compose the full return.

**Proposition 3.** *Let* $Q_n^\pi(s_t, a_t) = \mathbf{E}_{t,\pi,\mathcal{M}}[\sum_{j=t}^{t+n-1} \gamma^{j-t} r_j]$ *be the truncated return and* $Q_{n:\infty}^\pi(s_t, a_t) = \mathbf{E}_{t,\pi,\mathcal{M}}[\gamma^n Q(s_{t+n}, a_{t+n})]$ *the shifted return. Then* $Q^\pi(s_t, a_t) = Q_n^\pi(s_t, a_t) + Q_{n:\infty}^\pi(s_t, a_t)$ *represents the full return, i.e.* $Q^\pi(s_t, a_t) = \mathbf{E}_{t,\pi,\mathcal{M}}[\sum_{j=t}^{\infty} \gamma^{j-t} r_j]$.

The proof of Proposition 3 is in the appendix. Thus, we can formalize the update of the full Q-function by:

$$
\begin{aligned}
Q(s_t, a_t) \leftarrow (1 - \alpha_Q) Q(s_t, a_t) + \alpha_Q(r_t + \gamma(Q_n(s_{t+1}, \arg\max_a Q(s_{t+1}, a)) \\
+ Q_{n:\infty}(s_{t+1}, \arg\max_a Q(s_{t+1}, a)))).
\end{aligned}
\tag{8}
$$

The incorporation of truncated returns breaks down the time scale of the long-term prediction by the Shifted Q-function. We call this algorithm *Composite Q-learning* (cf. Algorithm 1). Please note that, in the tabular setting, Composite Q-learning is equivalent to Q-learning with learning rate $\alpha_Q$ when setting $\alpha_{\text{Tr}} = \alpha_{\text{Sh}} = \alpha_Q$. However, a notable advantage is that Composite Q-learning offers an independent optimization for the different learning rates corresponding to different temporal horizons.

---

**Algorithm 1:** Composite Q-learning

---

**1** initialize Truncated Q-functions $Q_i$

**2** initialize Shifted Q-functions $Q_{i:\infty}$

**3** initialize Q-function $Q$

**4** **for** *episode* $= 1..E$ **do**

**5**      get initial state $s_1$

**6**      **for** $t = 1..T$ **do**

**7**          apply $\epsilon$-greedy action $a_t$

**8**          observe $s_{t+1}$ and $r_t$

**9**          update Truncated Q-functions by:

            $Q_1(s_t, a_t) \leftarrow (1 - \alpha_{\text{Tr}}) Q_1(s_t, a_t) + \alpha_{\text{Tr}} r_t$

            $Q_{i>1}(s_t, a_t) \leftarrow (1 - \alpha_{\text{Tr}}) Q_i(s_t, a_t) + \alpha_{\text{Tr}}(r_t + \gamma Q_{i-1}(s_{t+1}, \arg\max_a Q(s_{t+1}, a)))$

         update Shifted Q-functions by:

            $Q_{1:\infty}(s_t, a_t) \leftarrow (1 - \alpha_{\text{Sh}}) Q_{1:\infty}(s_t, a_t) + \alpha_{\text{Sh}}(\gamma \max_a Q(s_{t+1}, a))$

            $Q_{(i>1):\infty}(s_t, a_t) \leftarrow (1 - \alpha_{\text{Sh}}) Q_{i:\infty}(s_t, a_t) + \alpha_{\text{Sh}}(\gamma Q_{(i-1):\infty}(s_{t+1}, \arg\max_a Q(s_{t+1}, a)))$

         update Q-function by:

            $Q(s_t, a_t) \leftarrow (1 - \alpha_Q) Q(s_t, a_t) + \alpha_Q(r_t + \gamma(Q_n(s_{t+1}, \arg\max_a Q(s_{t+1}, a))$

                       $+ Q_{n:\infty}(s_{t+1}, \arg\max_a Q(s_{t+1}, a))))$

---

### 3.2 Deep Composite Q-learning

In order to cope with infinite or continuous state and action-spaces, we extend Composite Q-learning to the function approximation setting.

#### 3.2.1 Target Formulation

Let $Q^{\mathrm{Tr}}(\cdot,\cdot|\theta^{\mathrm{Tr}})$ denote a function approximator with parameters $\theta^{\mathrm{Tr}}$ and $n$ outputs, subsequently called *heads*, estimating Truncated Q-functions $Q_i^*$. Each output $Q_i^{\mathrm{Tr}}$ bootstraps from the prediction of the preceding head, with the first approximating the immediate reward function. The targets are therefore given by:

$$y_{j,1}^{\mathrm{Tr}} = r_j \text{ and } y_{j,i>1}^{\mathrm{Tr}} = r_j + \gamma Q_{i-1}^{\mathrm{Tr}\prime}(s_{j+1}, \mu'(s_{j+1}|\theta^{\mu\prime})|\theta^{\mathrm{Tr}\prime}), \tag{9}$$

where $\mu'$ corresponds to the target-actor maximizing the full Q-value as defined in Section 2 and $Q_{i-1}^{\mathrm{Tr}\prime}$ the output of the respective Q-target-network. That is, $Q^{\mathrm{Tr}}$ represents evaluations of $\mu$ at different stages of truncation and $y_{j,i<n}^{\mathrm{Tr}}$ serve as intermediate predictions to get $y_{j,n}^{\mathrm{Tr}}$. We then *only* use $Q_n^{\mathrm{Tr}}$, which implements the full $n$-step return, as the first part of the composition of the Q-target, but not the $Q_{i<n}^{\mathrm{Tr}}$ values.

The second part of the composition is represented by the Shifted Q-function. Let $Q^{\mathrm{Sh}}(\cdot,\cdot|\theta^{\mathrm{Sh}})$ denote the function approximator estimating the Shifted Q-functions $Q_{i:\infty}^{\pi}$ by $n$ different outputs, parameterized by $\theta^{\mathrm{Sh}}$. We can shift the Q-prediction by bootstrapping without taking the immediate reward into account, so as to skip the first $n$ rewards of a target-policy rollout. The Shifted Q-targets for heads $Q_i^{\mathrm{Sh}}$ therefore become:

$$y_{j,1}^{\mathrm{Sh}} = \gamma Q'(s_{j+1}, \mu'(s_{j+1}|\theta^{\mu\prime})|\theta^{Q\prime}) \text{ and } y_{j,i>1}^{\mathrm{Sh}} = \gamma Q_{i-1}^{\mathrm{Sh}\prime}(s_{j+1}, \mu'(s_{j+1}|\theta^{\mu\prime})|\theta^{\mathrm{Sh}\prime}). \tag{10}$$

That is, each $Q_{i-1}^{\mathrm{Sh}\prime}$ implicitly estimates the $\gamma^{i-1}$ multiplier. In the function approximation setting, we can thus define the Composite Q-target as:

$$y_j^Q = r_j + \gamma(Q_n^{\mathrm{Tr}\prime}(s_{j+1}, \mu'(s_{j+1}|\theta^{\mu\prime})|\theta^{\mathrm{Tr}\prime}) + Q_n^{\mathrm{Sh}\prime}(s_{j+1}, \mu'(s_{j+1}|\theta^{\mu\prime})|\theta^{\mathrm{Sh}\prime})), \tag{11}$$

approximated by $Q(\cdot,\cdot|\theta^Q)$ with parameters $\theta^Q$. We jointly estimate $Q^{\mathrm{Tr}}$, $Q^{\mathrm{Sh}}$ and $Q$ with function approximator $Q^{\mathcal{C}}(\cdot,\cdot|\theta^{\mathcal{C}})$.

#### 3.2.2 Entropy Regularization

Each pair $Q_i^{\mathrm{Tr}} + Q_i^{\mathrm{Sh}}|_{1\leq i\leq n}$ is a complete approximation of the true Q-value. Note however, that it is also bootstrapped with the full Q-value. The circular dependency can lead to stability issues, due to the amplification of propagated errors. Additionally, higher learning rates for the Shifted Value-functions, as motivated in Section 3.1.2, may lead to overfitting. In particular, errors in the truncated predictions are propagated quickly by the Shifted Q-functions, especially when trained with a high learning rate. Therefore, it is important to keep the truncated predictions close across the different time steps while also preventing the Shifted Q-functions from running into suboptimal local minima. As a means to alleviate these issues, we add a regularization based on the entropy of a Gaussian distribution formed by mean $\mu(Q_i^{\mathrm{Tr}} + Q_i^{\mathrm{Sh}}|_{1\leq i\leq n})$ and variance $\sigma^2(Q_i^{\mathrm{Tr}} + Q_i^{\mathrm{Sh}}|_{1\leq i\leq n})$ over all $n$ complete predictions of $Q$. We can then formalize an incentive of the Truncated Q-functions to stay close between predictions whilst forcing the Shifted Q-functions to keep the distribution as broad as possible. We define the Gaussian distribution $\mathcal{N}_Q$ over all Q-predictions therefore by:

$$\mathcal{N}_Q\left(\mu(Q_i^{\mathrm{Tr}} + Q_i^{\mathrm{Sh}}|_{1\leq i\leq n}), \sigma^2(Q_i^{\mathrm{Tr}} + Q_i^{\mathrm{Sh}}|_{1\leq i\leq n})\right), \tag{12}$$

with corresponding entropy:

$$\mathcal{H}_Q = \frac{1}{2}\log\left(2\pi e\sigma^2(Q_i^{\mathrm{Tr}} + Q_i^{\mathrm{Sh}}|_{1\leq i\leq n})\right). \tag{13}$$

In order to enhance stability of the learning process as a whole while preventing the Shifted Q-functions from overfitting, we not only minimize the mean squared error between targets $y_j^{\mathcal{C}}$ and predictions $Q^{\mathcal{C}}(s_j, a_j|\theta^{\mathcal{C}})$,

but apply gradient descent on the entropy for the parameters of $Q^{\mathrm{Tr}}$ and gradient ascent on the parameters of $Q^{\mathrm{Sh}}$. More specifically, we define the squared error for some sample $j$ as:

$$\delta_j = \left(y_j^{\mathcal{C}} - Q^{\mathcal{C}}(s_j, a_j|\theta^{\mathcal{C}})\right)^2, \tag{14}$$

and the gradient with regard to all parameters of $Q^{\mathcal{C}}$ including the different learning rates as:

$$\xi_j = \alpha_Q \delta_j \nabla_{\theta^Q} Q^{\mathcal{C}}(s_j, a_j|\theta^{\mathcal{C}}) + \alpha_{\mathrm{Tr}} \delta_j \nabla_{\theta^{\mathrm{Tr}}} Q^{\mathcal{C}}(s_j, a_j|\theta^{\mathcal{C}}) + \alpha_{\mathrm{Sh}} \delta_j \nabla_{\theta^{\mathrm{Sh}}} Q^{\mathcal{C}}(s_j, a_j|\theta^{\mathcal{C}}). \tag{15}$$

The regularization then adds to the gradient:

$$\eta_j = \mathcal{H}(s_j, a_j)(\alpha_{\mathrm{Sh}}\beta_{\mathrm{Sh}}\nabla_{\theta^{\mathrm{Sh}}} - \alpha_{\mathrm{Tr}}\beta_{\mathrm{Tr}}\nabla_{\theta^{\mathrm{Tr}}}). \tag{16}$$

Hence, the parameter update in its simplest form becomes:

$$\theta^{\mathcal{C}} \leftarrow \theta^{\mathcal{C}} + \frac{1}{m}\sum_j (\xi_j - \eta_j). \tag{17}$$

However, a more sophisticated optimizer such as Adam (Kingma & Ba, 2015) can be applied analogously. A detailed description of Deep Deterministic Continuous Composite Q-learning is given in Algorithm 2. For an exemplary application within TD3 (as in our experiments in the following), Gaussian policy smoothing has to be added to all targets in Line 8, as well as taking the minimum prediction of two distinct critics for each target. Furthermore, actor and target networks have to be updated with delay.

---

**Algorithm 2:** Deep Deterministic Continuous Composite Q-learning

---

**1** initialize critic $Q^{\mathcal{C}}$, actor $\mu$ and targets $Q^{\mathcal{C}\prime}$, $\mu'$

**2** initialize replay buffer $\mathcal{R}$

**3** **for** $episode = 1..E$ **do**

**4**     get initial state $s_1$

**5**     **for** $t = 1..T$ **do**

**6**        apply action $a_t = \mu(s_t|\theta^\mu) + \xi$, where $\xi \sim \mathcal{N}(0, \sigma)$

**7**        observe $s_{t+1}$ and $r_t$ and save transition $(s_t, a_t, s_{t+1}, r_t)$ in $\mathcal{R}$

**8**        calculate targets:

$$y_{j,1}^{\mathrm{Tr}} = r_j$$
$$y_{j,i>1}^{\mathrm{Tr}} = r_j + \gamma Q_{i-1}^{\mathrm{Tr}\prime}(s_{j+1}, \mu'(s_{j+1}|\theta^{\mu\prime})|\theta^{\mathrm{Tr}\prime})$$
$$y_{j,1}^{\mathrm{Sh}} = \gamma Q'(s_{j+1}, \mu'(s_{j+1}|\theta^{\mu\prime})|\theta^{Q\prime})$$
$$y_{j,i>1}^{\mathrm{Sh}} = \gamma Q_{i-1}^{\mathrm{Sh}\prime}(s_{j+1}, \mu'(s_{j+1}|\theta^{\mu\prime})|\theta^{\mathrm{Sh}\prime})$$
$$y_j^Q = r_j + \gamma(Q_n^{\mathrm{Tr}\prime}(s_{j+1}, \mu'(s_{j+1}|\theta^{\mu\prime})|\theta^{\mathrm{Tr}\prime}) + Q_n^{\mathrm{Sh}\prime}(s_{j+1}, \mu'(s_{j+1}|\theta^{\mu\prime})|\theta^{\mathrm{Sh}\prime}))$$
$$y_j^{\mathcal{C}} = \left[y_j^Q, y_{j,1}^{\mathrm{Tr}}, y_{j,i>1}^{\mathrm{Tr}}, y_{j,1}^{\mathrm{Sh}}, y_{j,i>1}^{\mathrm{Sh}}\right]$$

**9**        update $Q^{\mathcal{C}}$ on minibatch $b$ of size $m$ from $\mathcal{R}$ according to Equation (17)

**10**        update $\mu$ on $Q$ based on the sampled deterministic policy gradient

**11**        adjust parameters of $Q^{\mathcal{C}\prime}$ and $\mu'$

---

### 3.3 Off-policy TD($\Delta$)

Another way to divide the value function into multiple time scales is TD($\Delta$) (Romoff et al., 2019). To this point, it has only been applied in an on-policy setting. In favor of comparability, we extend TD($\Delta$) to Q-learning, yielding TD3($\Delta$). The main idea of TD($\Delta$) is the combination of different value functions corresponding to increasing discount values. Let $\gamma^\Delta$ denote a fixed ordered sequence of increasing discount values:

$$\gamma^\Delta = (\gamma_1, \gamma_2, \ldots, \gamma_k)^\top |_{\gamma_i > \gamma_{i-1}}. \tag{18}$$

We can then define delta functions $W_i$ as:

$$W_1 = Q_{\gamma_1} \text{ and } W_{i>1} = Q_{\gamma_i} - Q_{\gamma_{i-1}}. \tag{19}$$

Let $Q^\Delta(\cdot, \cdot | \theta^\Delta)$ denote the function approximator estimating $Q_{\gamma_{1 \le i \le k}}$ with parameters $\theta^\Delta$. Based on the derivations in (Romoff et al., 2019), the targets for Q-learning can be formalized as:

$$y_{j,1}^\gamma = r_j + \gamma_1 Q_{\gamma_1}'(s_{j+1}, \mu'(s_{j+1} | \theta^{\mu'}) | \theta^{\Delta'}) \text{ and}$$
$$y_{j,i>1}^\gamma = (\gamma_i - \gamma_{i-1}) Q_{\gamma_{i-1}}'(s_{j+1}, \mu'(s_{j+1} | \theta^{\mu'}) | \theta^{\Delta'}) + \gamma_i W_i'(s_{j+1}, \mu'(s_{j+1} | \theta^{\mu'}) | \theta^{\Delta'}). \tag{20}$$

The authors suggest the use of $n$-step targets within TD($\Delta$) which is not easily applicable in an off-policy setting. Furthermore, Composite Q-learning is based on *single-step* transitions. To keep the comparison fair and clear, we thus compare our approach to single-step Off-policy TD($\Delta$) in our experiments. A detailed overview of Deep Deterministic Continuous Off-policy TD($\Delta$) can be found in Algorithm 3. To transform Off-policy TD($\Delta$) to TD3($\Delta$), the adjustments as described in Section 3.2.2 have to be applied analogously.

---

**Algorithm 3:** Deep Deterministic Continuous Off-policy TD($\Delta$)

**1** initialize critic $Q^\Delta$, actor $\mu$ and targets $Q^{\Delta'}$, $\mu'$

**2** initialize replay buffer $\mathcal{R}$

**3** set discount values $\gamma^\Delta = (\gamma_0, \gamma_1, \ldots, \gamma_k)^\top$

**4** **for** *episode* $= 1..E$ **do**

**5**      get initial state $s_1$

**6**      **for** $t = 1..T$ **do**

**7**          apply action $a_t = \mu(s_t | \theta^\mu) + \xi$, where $\xi \sim \mathcal{N}(0, \sigma)$

**8**          observe $s_{t+1}$ and $r_t$ and save transition $(s_t, a_t, s_{t+1}, r_t)$ in $\mathcal{R}$

**9**          calculate targets:

$$y_{j,1}^\gamma = r_j + \gamma_1 Q_{\gamma_1}'(s_{j+1}, \mu'(s_{j+1} | \theta^{\mu'}) | \theta^{\Delta'})$$
$$y_{j,i>1}^\gamma = (\gamma_i - \gamma_{i-1}) Q_{\gamma_{i-1}}'(s_{j+1}, \mu'(s_{j+1} | \theta^{\mu'}) | \theta^{\Delta'})$$
$$+ \gamma_i W_i'(s_{j+1}, \mu'(s_{j+1} | \theta^{\mu'}) | \theta^{\Delta'})$$

**10**          update $Q^\Delta$ on minibatch $b$ of size $m$ from $\mathcal{R}$

**11**          update $\mu$ on $Q_{\gamma_k}$ based on the sampled deterministic policy gradient

**12**          adjust parameters of $Q^{\Delta'}$ and $\mu'$

---

# 4 Experimental Results

In this section, we evaluate Tabular and Deep Composite Q-learning. In the tabular case, we apply Composite Q-learning on deterministic and stochastic chain MDPs of varying horizon to illustrate and analyze the characteristics of the interplay between short-term and long-term predictions. We then discuss advantages of Deep Composite Q-learning under a noise-free reward for the two robot simulation tasks Walker2d-v2 and Hopper-v2 and evaluate Deep Composite Q-learning for Walker2d-v2, Hopper-v2 and Humanoid-v2 under a noisy reward function, comparing it to TD3 and TD3($\Delta$).

## 4.1 Tabular Composite Q-learning

### 4.1.1 Deterministic Chain

We evaluate the effect of composing the long-term return of multiple short-term predictions in the tabular setting. We apply Composite Q-learning in the tabular case to the MDP of horizon $K$ given in Figure 2a. We compare Composite Q-learning to vanilla Q-learning, as well as multi-step Q-learning based on subtrajectories of the exploratory policy and imaginary rollouts of the target-policy with the true model of the MDP as a hypothetical lower bound on the required updates.

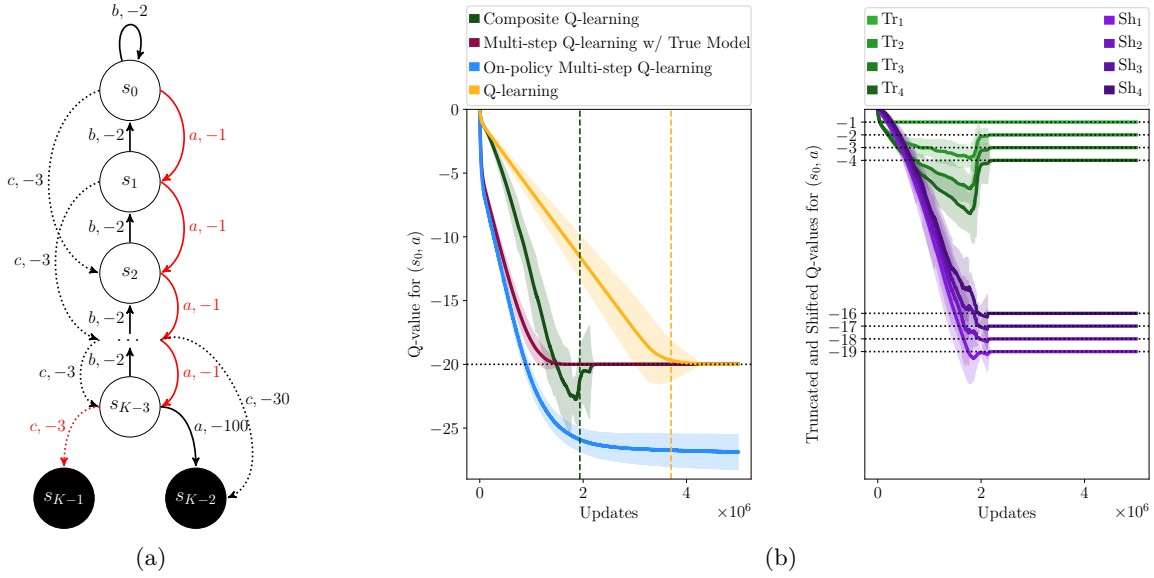

(a)            (b)

Figure 2: (a) In this deterministic-chain MDP of horizon $K$, the agent ought to arrive at terminal state $s_{K-1}$ using actions $\{a, b, c\}$. The initial state is $s_0$ and the optimal policy is given in red. (b) Mean results and two standard deviations over 10 runs on the MDP with a horizon of $K = 20$. The left plot depicts the value of $s_0$ and action $a$ as estimated by the different approaches over time. Dashed lines indicate convergence to the optimal policy. The predicted Truncated Q-values for state $s_0$ and action $a$ with horizons 1 to 4, denoted by $\mathrm{Tr}_1, \ldots, \mathrm{Tr}_4$, and predicted Shifted Q-values for state $s_0$ and action $a$, denoted by $\mathrm{Sh}_1, \ldots, \mathrm{Sh}_4$, are to the right. Dotted lines indicate the true optimal respective Q-values.

Results for $K = 20$ are depicted in Figure 2b, where mean performances of the approaches are shown on the left and the intermediate predictions of Composite Q-learning on the right. All approaches update the Q-function with a learning rate of $10^{-3}$ on the same fixed batch of $10^3$ episodes with a percentage of 10% non-optimal transitions. We set rollout length $n = 4$ for all respective approaches. Please note that this value can be set arbitrarily, however in order to be beneficial it should be set to a value smaller than the temporal horizon of the task. In this example, a rollout length of 4 corresponds to the integer square root of the horizon of the MDP, meaning that Truncated and Shifted Q-functions have a similar temporal horizon

Table 1: Comparison of convergence speed between Tabular Q-learning and Tabular Composite Q-learning for exemplary runs on the MDP given in Figure 2a with $n = 4$.

| Horizon $K$ | 10 | 20 | 50 | 100 |
|---|---|---|---|---|
| Speed up over Q-learning | 11% | 44% | 57% | 66% |

due to the partitioning of the long-term value. An evaluation of the performance for different rollout lengths $n$ is shown in Figure B.1a in the appendix.

For the experiments in Figure 2b, we update the Shifted Q-function with a learning rate of $10^{-2}$ and the Truncated Q-functions with a learning rate of $10^{-3}$. Composite Q-learning converges much faster to the true action-value than vanilla Q-learning and is close to the multi-step approach based on model rollouts (hypothetical lower bound). The erroneous updates of on-policy multi-step Q-learning lead to convergence to a wrong action-value which is underlining the importance of truly off-policy learning.

Composite Q-learning estimates the short-term rollouts off-policy by design and shifts the long-term value in time so as to have a consistent temporal chaining. All intermediate predictions therefore converge to the true intermediate action-values, which is shown on the right side of Figure 2b. The difference in convergence speed between Q-learning and Composite Q-learning grows with increasing horizon, as shown in Table 1 for the described hyperparameter setting.

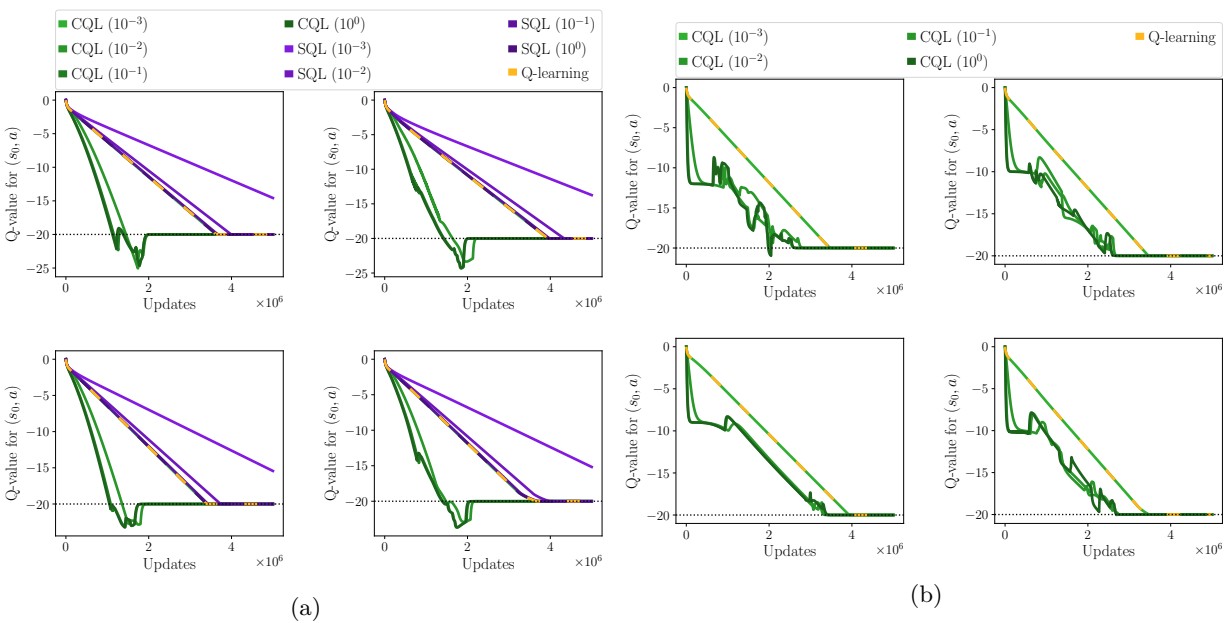

Figure 3: Results of 4 individual runs on the deterministic chain MDP with a horizon of $K = 20$ for Composite and Shifted Q-learning, (a) with different learning rates for the Shifted Q-function and (b) with different learning rates for the Truncated Q-function (denoted by the numbers in parentheses). The learning rates for the full Q-function and for the (a) Truncated Q-functions and the (b) Shifted Q-functions are set to $10^{-3}$ in all experiments.

Next, we present an evaluation of different learning rates for the Shifted Q-functions, depicted in Figure 3a. Here we compare Composite Q-learning (CQL) and vanilla Q-learning to Shifted Q-learning (SQL), which corresponds to Q-learning with a one-step shifted target (replacing the bootstrap of the future return with an evaluation of a one-step Shifted Q-function and without approximate $n$-step returns from a Truncated Q-function). The results show that shifting the value in time alone is slowing down convergence. Precisely,

Shifted Q-learning with a learning rate of 1.0 is equivalent to vanilla Q-learning; the same holds for Composite Q-learning with a learning rate of $10^{-3}$ for the Shifted Q-function.

Please note that although the learning rate of Q-learning could be set to a higher value for this very MDP, this example shows the benefit of Composite Q-learning even though higher learning rates for Shifted Q-functions alone do not bring any speedup in convergence. Only the combination of Shifted Q-functions and Truncated Q-functions lead to an increase of convergence speed, even in the case when Truncated and full Q-function are estimated with the same (slow) learning rate of $10^{-3}$. The results underline that the higher learning rates for the Shifted Q-functions only have a beneficial effect in combination with truncated predictions.

The counterpart with different learning rates for the Truncated Q-functions, keeping the learning rates for the full Q-estimate and Shifted Q-functions fixed, can be seen in Figure 3b. While there is improvement in convergence using a larger learning rate for the Truncated Q-functions, the results show higher variance and less benefit than the Shifted Q-functions in Figure 3a.

### 4.1.2   Stochastic Chain

We further investigate the effect of stochastic immediate rewards on Composite Q-learning on the chain MDP shown in Figure 4. All actions lead further into the chain, however they yield different expected rewards. Action $a$ has an expected reward of $-0.8$, corresponding to an immediate reward of $-1$ with 80% chance and 0 with 20%. Action $b$, on the other hand, has an expected reward of $-1.01$, however it is $+1$ with 99% chance and $-200$ with 1%. Composite Q-learning with learning rates of $10^{-2}$ for the full Q-function, $10^{-1}$ for the Shifted Q-functions and $10^{-3}$ for the Truncated Q-functions is compared to Q-learning with the same respective learning rates.

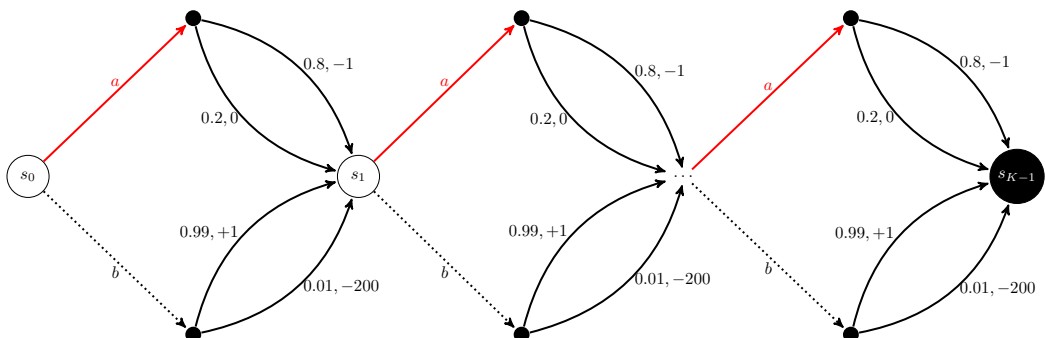

Figure 4: In this stochastic chain MDP of horizon $K$, the agent ought to arrive at terminal state $s_{K-1}$ using actions $a$ and $b$. The initial state is $s_0$. Transitions are stochastic. The optimal policy is given in red.

Results for different horizons and no discount are depicted in Figure 5. It can be seen that Q-learning with a high learning rate is converging to a wrong action-value and the deviation becomes more significant the longer the horizon of the task. The same holds for Q-learning with a learning rate of $10^{-2}$, however the deviation becomes smaller. Moreover, Q-learning with the same learning rate as the full Q-function in Composite Q-learning is converging much slower for growing horizons, yet the learning rate for the Truncated Q-functions is 10 times smaller than for Q-learning. This emphasizes the mutual interplay of Truncated and Shifted Q-functions. Furthermore, Composite Q-learning is converging to the true optimal action-value. Q-learning with a small learning rate of $10^{-3}$ is far from achieving a satisfying level of convergence within the time frame of $2 \cdot 10^7$ updates.

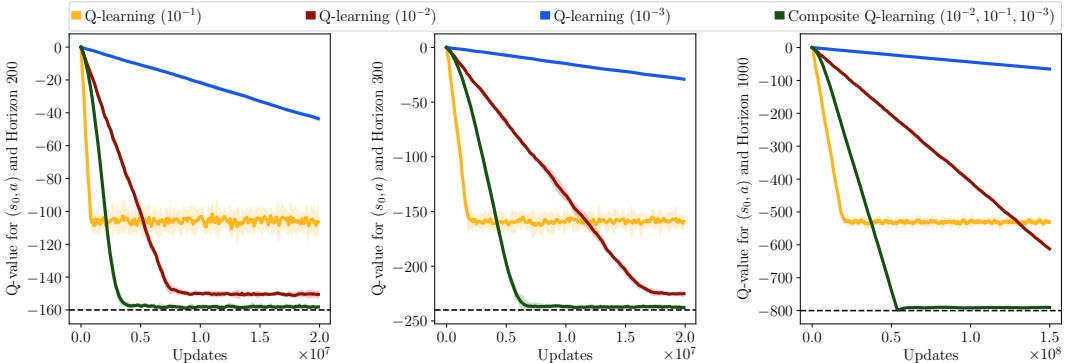

Figure 5: Results of Composite Q-learning and Q-learning with different learning rates over 5 training runs for horizons (left) 200, (middle) 300 and (right) 1000 and no discount. The true action-value is indicated by the dashed line. Composite Q-learning uses a learning rate of $10^{-2}$ for the full Q-value, $10^{-1}$ for the shifted Q-values and $10^{-3}$ for the truncated Q-values within the same experimental evaluation.

Shifted Q-functions thus allow for higher learning rates which in combination with a more cautious fitting of short-term predictions leads to increased data efficiency while achieving a smaller deviation from the true action-value. Most importantly, the difference in performance between Q-learning and Composite Q-learning (better value estimation with respect to to a high learning rate and better convergence properties comparing to a small learning rate) becomes larger for longer horizons. The performance for different rollout lengths are shown in Figure B.1b in the appendix.

## 4.2 Composite Q-learning with Function Approximation

We apply Composite Q-learning within TD3 and compare against TD3 and TD3($\Delta$) on three robot simulation tasks of OpenAI Gym (Brockman et al., 2016) based on MuJoCo (Todorov et al., 2012): Walker2d-v2, Hopper-v2 and Humanoid-v2. A visualization of the environments is depicted in Figure 6. We discuss hyperparameter optimization and hyperparameter settings in Appendix C. We analyze the properties of Composite Q-learning in terms of data-efficiency and stability on the true reward for Walker2d-v2 and the potential increase of updates per sample for Walker2d-v2 and Hopper-v2 in Section 4.2.1[3], before we evaluate Composite TD3 on a very noisy reward signal for Walker2d-v2, Hopper-v2 and Humanoid-v2 in Section 4.2.2.[4] As main evaluation metric for performance and stability, we provide the mean area under the learning curve normalized by the mean area of Composite Q-learning, in order to set the mean performance of Composite Q-learning as the origin for comparison.



Figure 6: Visualization of Walker2d-v2 (left), Hopper-v2 (middle) and Humanoid-v2 (right).

### 4.2.1 True Reward Function

In a first experiment, we compare Composite TD3, TD3($\Delta$) and conventional TD3 with both the optimized default learning rate and the highest learning rate used in Composite TD3 (TD3 HL) on the true reward for the Walker2d-v2 environment in Figure 7a over 8 training runs. Composite TD3 and TD3($\Delta$) yield a better and more stable performance than TD3 with either learning rate, with Composite TD3 being

---

[3]An additional comparison to two multi-step baselines is in Appendix F.

[4]An additional evaluation of Composite Q-learning within DQN (Mnih et al., 2015) on the LunarLander-v2 environment with noisy rewards can be found in Appendix E.

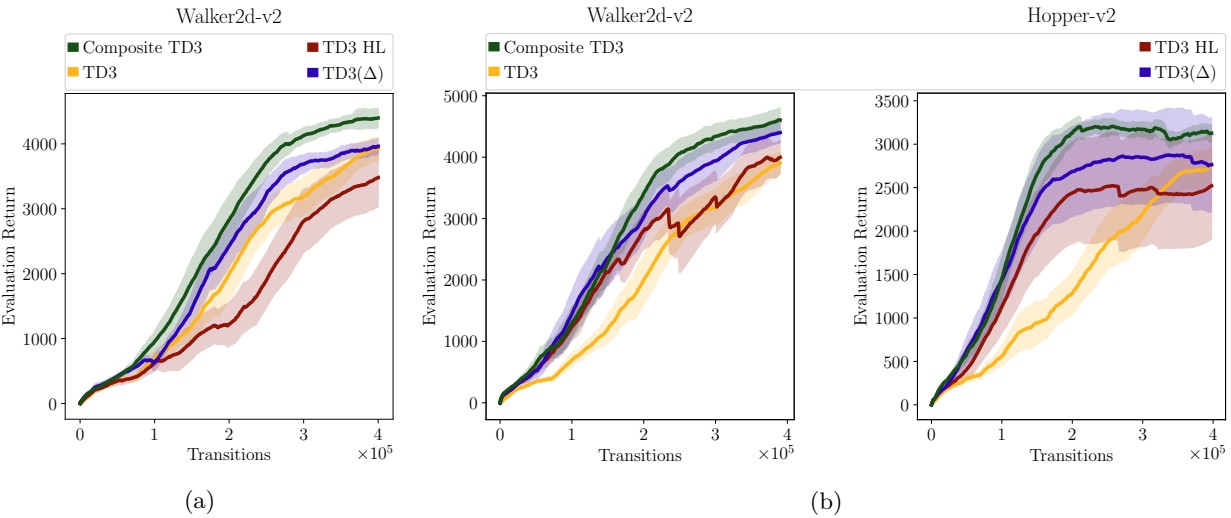

Figure 7: (a) Mean performance and half a SD (standard deviation) over 8 training runs for the Walker2d-v2 environment with the default reward. (b) Mean performance and half a SD over 8 training runs of Composite Q-learning on the vanilla reward function and multiple update steps per collected sample for the Walker2d-v2 and Hopper-v2 environment.

the best performing of all approaches. The differences between conventional TD3 and Composite TD3 are significant, as shown in Table D.1 in the appendix. TD3 is not able to gain performance with higher learning rate and even degenerates. Composite TD3, on the other hand, *can* take advantage from the higher learning rate. We hence argue that the extension of TD3 to Composite TD3 allows for higher learning rates compared to the vanilla version, at least for some of the individual parts of value-estimation. In addition, Composite TD3 and TD3($\Delta$) show less variance.

In a further experiment using the true reward, we investigate the potential increase in data-efficiency when applying multiple gradient steps per collected sample. Results over 8 training runs can be seen in Figure 7b, with corresponding significance tests provided in Table D.1 in the appendix. While there seems to be a limit in benefit for TD3 with respect to multiple gradient steps, Composite TD3 and TD3($\Delta$) yield a faster learning speed. In addition, Composite Q-learning has no increase in variance opposed to all other approaches. The high learning rate in combination with multiple gradient steps is harmful with respect to learning stability.

The influence of the horizon of Truncated and Shifted Q-functions, as well as the respective regularization weights maximizing and minimizing the entropy, is exemplified for the Walker2d-v2 environment in Figure 8. It can be seen that the regularization of the Shifted Q-functions has a higher influence on the performance. If the weight is set too high, the variance increases significantly. The same holds for a temporal horizon that is too long. The regularization of the Truncated Q-functions has less influence on convergence and variance, however, please note the strong connection of these hyperparameters. Therefore, the best triad setting needs to be found for a given problem and yields an interesting direction for meta-reinforcement learning. Prior experiments with other regularization terms only minimizing the entropy also showed enhanced stability, however, it is important to prevent the Shifted Q-functions from overfitting when applying higher learning rates in the TD3 setting. It can be seen that there is a trade-off between complexity of truncated value-estimation and the faster composition of the value by the Shifted Q-functions. If the rewards are very noisy, it appears to be of advantage to set $n$ to a smaller value in order to simplify the learning problem of truncated value-estimation (as in our experiments in Section 4.2.2). On the other hand, in order to make use of the temporal chaining of the Shifted Q-functions, the horizon should cover multiple time steps.

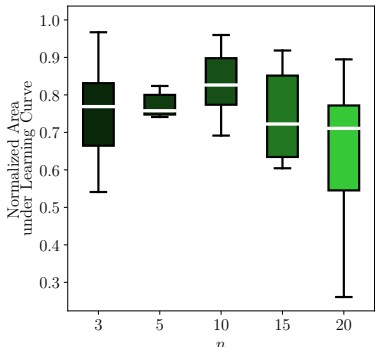 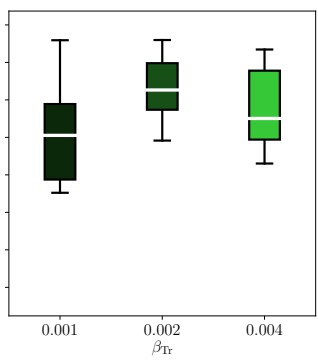 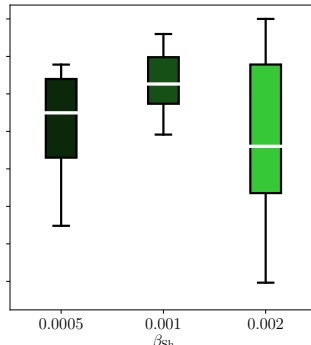

Figure 8: Normalized area under the learning curve for Composite TD3 in the Walker2d-v2 environment with different truncation horizons $n$ (left) and different regularization weights $\beta^{\mathrm{Tr}}$ (middle) and $\beta^{\mathrm{Sh}}$ (right). The plots show median and interquartile ranges over 8 training runs, each representing mean evaluation performance over 100 initial states.

### 4.2.2 Noisy Reward Function

We evaluate Composite TD3 under a noisy reward function and compare to TD3, TD3 with high learning rate and TD3($\Delta$). The immediate reward is replaced by a uniform sample $u \sim \mathcal{U}[-1, 1]$ with 40% chance, corresponding to the strongest noise level described in (Romoff et al., 2019). In order to account for the high variance, we provide mean and standard deviation (SD) over 8 runs. Results can be seen in Figure 9, corresponding significance tests are provided in Table D.2 in the appendix. Composite TD3 proves to be very robust, even for the very complex Humanoid-v2 environment. All other approaches suffer from slower learning speed and high variance. This holds especially for TD3 with high learning rate and TD3($\Delta$). We believe this to be caused by overfitting to the noisy reward function due to the high learning rate in TD3 HL and due to the low gamma of the first value functions in TD3($\Delta$). Since all other predictions only add to the prediction of the preceding head, there is nothing to prevent the overestimation bias from being propagated to the other outputs of higher discount value. In Composite Q-learning however, Shifted and Truncated Q-functions estimate distinct parts of the temporal chain. This is also highlighted in Figure 10, where the TD-errors of Shifted and Truncated Q-functions can be seen along with learning curves of Humanoid-v2. Whilst in the default setting with vanilla reward the Truncated Q-functions are converging much faster than the Shifted Q-functions due to the simpler problem induced by the smaller horizon, the opposite holds in

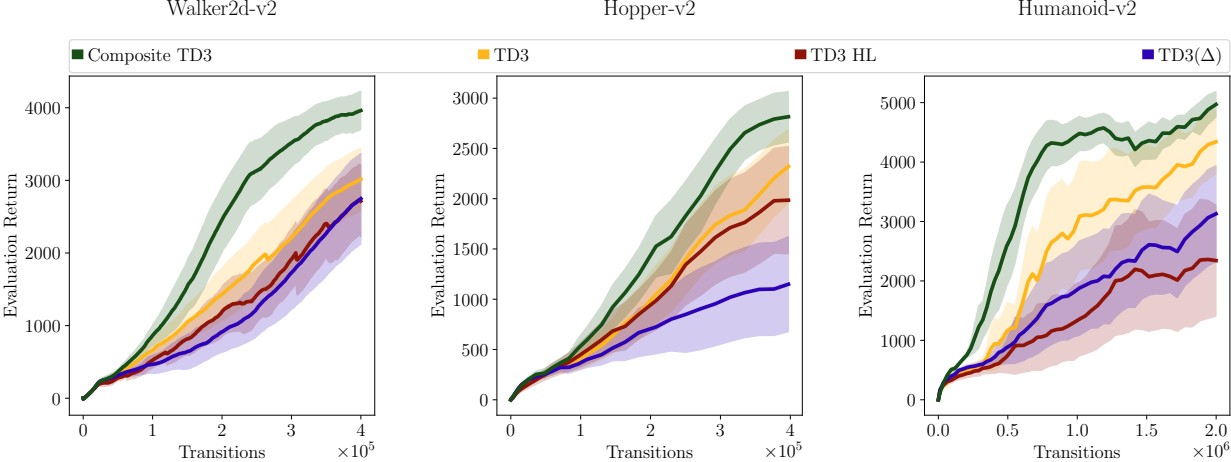

Figure 9: Mean performance and half a SD over 8 training runs for (left) Walker2d-v2, (middle) Hopper-v2 and (right) Humanoid-v2 with uniform noise on the reward function as in (Romoff et al., 2019).

Table 2: Mean normalized area under the learning curve and SD over 8 training runs of Composite TD3 in the noisy reward experiments.

| Method | Walker2d-v2 | Hopper-v2 | Humanoid-v2 |
|---|---|---|---|
| TD3 | $68\% \pm 22\%$ | $76\% \pm 28\%$ | $68\% \pm 30\%$ |
| TD3 (HL) | $56\% \pm 18\%$ | $72\% \pm 37\%$ | $38\% \pm 29\%$ |
| TD3($\Delta$) | $52\% \pm 23\%$ | $49\% \pm 31\%$ | $56\% \pm 26\%$ |
| Composite TD3 | $100\% \pm 20\%$ | $100\% \pm 25\%$ | $100\% \pm 16\%$ |

case of the noisy reward. In these experiments, the Truncated Q-functions are not capable of an accurate prediction within the given time frame, yet the Shifted Q-functions can construct an accurate chain given those inaccurate predictions reliably leading to a well-performing policy. This backs our hypothesis that Shifted Q-functions can benefit from higher learning rates also in noisy reward settings whereas Truncated Q-functions have to account for variance in the reward as well, which adds to the complexity of the problem significantly. This translates to faster learning, as underlined by the results in Table 2. It can furthermore be observed in Figure 10 that Truncated Q-functions corresponding to shorter horizons yield a smaller TD-error than Truncated Q-functions corresponding to longer horizons. The same holds for Shifted Q-functions where estimations at a later point in time (that is, with stronger shifting) have a lower TD-error than Shifted Q-functions estimating from an earlier point in time. This also highlights the importance of hyperparameter $n$, since it directly influences the point in time at which Shifted and Truncated Q-functions are combined. Please note that due to the recursive dependencies of the composite action-value and the – possibly very long – horizon of a task, also small differences can propagate quite readily.

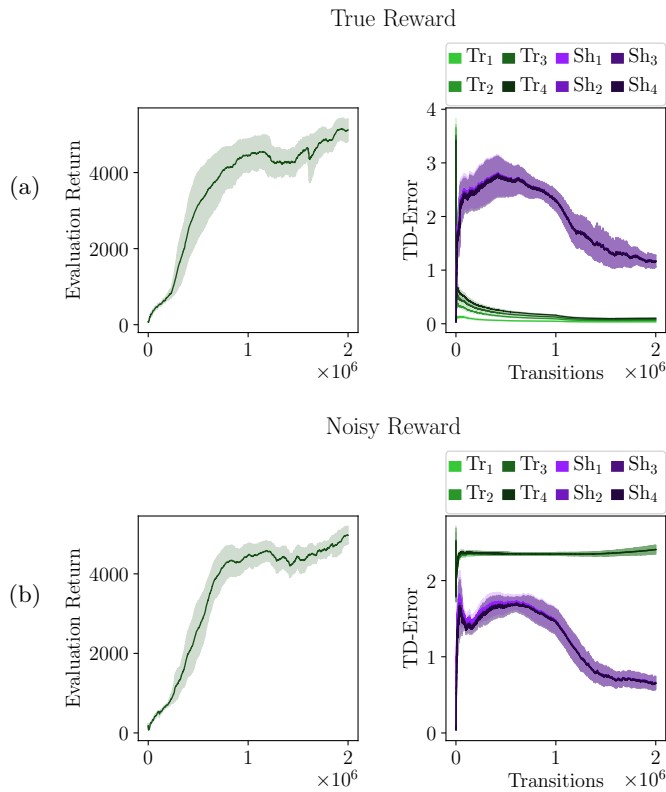

Figure 10: Performance and TD-errors on Humanoid-v2 of Truncated and Shifted Q-functions at $n = 1, \ldots, 4$ with respect to the (a) vanilla and (b) noisy reward. Please note that *TD-error* here means the deviation from the associated target.

Table 3: Mean maximum return and SD over 8 training runs for the noisy reward experiments.

| Method | Walker2d-v2 | Hopper-v2 | Humanoid-v2 |
|---|---|---|---|
| TD3 | $3063 \pm 813$ | $2386 \pm 729$ | $4453 \pm 1070$ |
| TD3 (HL) | $2950 \pm 866$ | $2124 \pm 1054$ | $2506 \pm 2004$ |
| TD3($\Delta$) | $2772 \pm 1245$ | $1242 \pm 954$ | $3251 \pm 1687$ |
| Composite TD3 | $4041 \pm 476$ | $2931 \pm 457$ | $5019 \pm 404$ |

Table 3 shows the mean maximum return achieved by Composite TD3, TD3 with default and high learning rate and TD3($\Delta$). Composite TD3 reaches significantly higher returns with lowest variance compared to vanilla TD3 when applied to a noisy reward function, especially for the most complex environment Humanoid-v2.

Our experimental results show that Composite Q-learning offers a significant improvement over traditional Q-learning methods and a great potential for real-world applications such as learning directly on real robots, where the extraction of the immediate reward signal can be noisy due to occlusions, complex training setups, sensor and actuator noise or fitted reward functions.

## 5   Conclusion

We introduced Composite Q-learning, an off-policy reinforcement learning method that divides learning of the full long-term value into a series of well-defined shorter prediction horizon estimates. It combines Truncated Q-functions acting on a short horizon with Shifted Q-functions for the remainder of the rollout. As a baseline, we further introduced and evaluated TD3($\Delta$), an off-policy variant of TD($\Delta$). We showed on three simulated robot learning tasks that compositional Q-learning methods can be of advantage with respect to data-efficiency compared to vanilla Q-learning methods and that Composite TD3 outperforms vanilla TD3 by 24% - 32% in terms of area under the learning curve. In addition, Composite Q-learning proved to be very robust to noisy reward signals which is very important for real-world applications where the immediate reward may depend on a stochastic next state or where the reward function is estimated from data.

Going forward, using ensemble uncertainty estimates based on the variance of all predictions from the Composite Q-function could be of benefit in update calculation, transition sampling and exploration. We further believe our method to be a better fit for non-stationary reward functions than traditional Q-learning methods due to the flexibility provided by the divided long-term return. Representing the farsighted return after $n$ steps of a policy rollout in the future, Shifted Q-functions could serve as an implicit dynamics model to further build the bridge between model-based and model-free methods. Lastly, the evaluation of further decomposition is a promising direction for future work and could bring even more flexibility to the learning process. It could be accomplished by shifting a truncated value-estimation in time. However, this would probably come at the cost of more hyperparameters, which could make the employment of more sophisticated hyperparameter optimization and meta-learning methods necessary.

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

# A  Theory of Composite Q-learning

## A.1  Proof of Proposition 1

First, we provide a theoretical foundation for Composite Q-learning.

**Proposition A.1.** *Let $Q_1^\pi(s_t, a_t) = r_t$ be the one-step Truncated Q-function and $Q_{i>1}^\pi(s_t, a_t) = r_t + \gamma \mathbf{E}_{t,\pi,\mathcal{M}}[Q_{i-1}^\pi(s_{t+1}, a_{t+1})]$ the i-step Truncated Q-function. Then $Q_i^\pi(s_t, a_t)$ represents the truncated return $Q_i^\pi(s_t, a_t) = \mathbf{E}_{t,\pi,\mathcal{M}}[\sum_{j=t}^{t+i-1} \gamma^{j-t} r_j].$*

*Proof.* Proof by induction. $Q_1^\pi(s_t, a_t) = r_t$ by definition. The theorem follows from induction step:

$$
\begin{aligned}
Q_i^\pi(s_t, a_t) &= r_t + \gamma \mathbf{E}_{\pi,\mathcal{M}} \left[ Q_{i-1}^\pi(s_{t+1}, a_{t+1}) \right] \\
&= r_t + \gamma \mathbf{E}_{\pi,\mathcal{M}} \left[ \sum_{j=(t+1)}^{(t+1)+(i-1)-1} \gamma^{j-(t+1)} r_j \right] \\
&= r_t + \gamma \mathbf{E}_{\pi,\mathcal{M}} \left[ \sum_{j=(t+1)}^{t+i-1} \gamma^{j-(t+1)} r_j \right] \\
&= r_t + \mathbf{E}_{\pi,\mathcal{M}} \left[ \sum_{j=(t+1)}^{t+i-1} \gamma^{j-t} r_j \right] \\
&= \mathbf{E}_{\pi,\mathcal{M}} \left[ \sum_{j=t}^{t+i-1} \gamma^{j-t} r_j \right].
\end{aligned}
$$

$\square$

## A.2  Proof of Proposition 2

**Proposition A.2.** *Let $Q_{1:\infty}^\pi(s_t, a_t) = \mathbf{E}_{t,\pi,\mathcal{M}}[\gamma Q^\pi(s_{t+1}, a_{t+1})]$ be the one-step Shifted Q-function and $Q_{i>1:\infty}^\pi(s_t, a_t) = \mathbf{E}_{t,\pi,\mathcal{M}}[\gamma Q_{i-1:\infty}^\pi(s_{t+1}, a_{t+1})]$ the i-step Shifted Q-function. Then $Q_{i:\infty}^\pi(s_t, a_t)$ represents the shifted return $Q_{i:\infty}^\pi(s_t, a_t) = \mathbf{E}_{t,\pi,\mathcal{M}}[\gamma^i Q^\pi(s_{t+i}, a_{t+i})].$*

*Proof.* Proof by induction. $Q_{1:\infty}^\pi(s_t, a_t) = \mathbf{E}_{\pi,\mathcal{M}}[\gamma Q^\pi(s_{t+1}, a_{t+1})]$ by definition. The theorem follows from induction step:

$$
\begin{aligned}
Q_{i:\infty}^\pi(s_t, a_t) &= \mathbf{E}_{\pi,\mathcal{M}} \left[ \gamma Q_{i-1:\infty}^\pi(s_{t+1}, a_{t+1}) \right] \\
&= \mathbf{E}_{\pi,\mathcal{M}} \left[ \gamma (\gamma^{i-1} Q^\pi(s_{t+1+i-1}, a_{t+1+i-1})) \right] \\
&= \mathbf{E}_{\pi,\mathcal{M}} \left[ \gamma (\gamma^{i-1} Q^\pi(s_{t+i}, a_{t+i})) \right] \\
&= \mathbf{E}_{\pi,\mathcal{M}} \left[ \gamma^i Q^\pi(s_{t+i}, a_{t+i}) \right].
\end{aligned}
$$

$\square$

## A.3  Proof of Proposition 3

**Proposition A.3.** *Let $Q_n^\pi(s_t, a_t) = \mathbf{E}_{t,\pi,\mathcal{M}}[\sum_{j=t}^{t+n-1} \gamma^{j-t} r_j]$ be the truncated return and $Q_{n:\infty}^\pi(s_t, a_t) = \mathbf{E}_{t,\pi,\mathcal{M}}[\gamma^n Q(s_{t+n}, a_{t+n})]$ the shifted return. Then $Q^\pi(s_t, a_t) = Q_n^\pi(s_t, a_t) + Q_{n:\infty}^\pi(s_t, a_t)$ represents the full return, i.e. $Q^\pi(s_t, a_t) = \mathbf{E}_{t,\pi,\mathcal{M}}[\sum_{j=t}^{\infty} \gamma^{j-t} r_j].$*

*Proof.*

$$Q^\pi(s_t, a_t) = Q_n^\pi(s_t, a_t) + Q_{n:\infty}^\pi(s_t, a_t)$$

$$= \mathbf{E}_{\pi,\mathcal{M}}\left[\sum_{j=t}^{t+n-1} \gamma^{j-t}r_j + \gamma^n Q^\pi(s_{t+n}, a_{t+n})\right]$$

$$= \mathbf{E}_{\pi,\mathcal{M}}\left[\sum_{j=t}^{t+n-1} \gamma^{j-t}r_j + \gamma^n \left(Q_n^\pi(s_{t+n}, a_{t+n}) + Q_{n:\infty}^\pi(s_{t+n}, a_{t+n})\right)\right]$$

$$= \mathbf{E}_{\pi,\mathcal{M}}\left[\sum_{j=t}^{t+n-1} \gamma^{j-t}r_j + \gamma^n \left(\sum_{j=t+n}^{t+2n-1} \gamma^{j-t-n}r_j + \gamma^n Q^\pi(s_{t+2n}, a_{t+2n})\right)\right]$$

$$= \mathbf{E}_{\pi,\mathcal{M}}\left[\sum_{j=t}^{t+n-1} \gamma^{j-t}r_j + \sum_{j=t+n}^{t+2n-1} \gamma^{j-t}r_j + \gamma^{2n} Q^\pi(s_{t+2n}, a_{t+2n})\right]$$

$$= \mathbf{E}_{\pi,\mathcal{M}}\left[\sum_{j=t}^{t+2n-1} \gamma^{j-t}r_j + \gamma^{2n} Q^\pi(s_{t+2n}, a_{t+2n})\right].$$

By induction, it follows $Q^\pi(s_t, a_t) = \mathbf{E}_{\pi,\mathcal{M}}[\sum_{j=t}^{\infty} \gamma^{j-t}r_j]$. $\qquad\square$

## B  Additional Evaluation of Tabular Composite Q-learning

In this section, we provide an additional evaluation of tabular Composite Q-learning applied to the MDPs in Figure 2a and Figure 4.

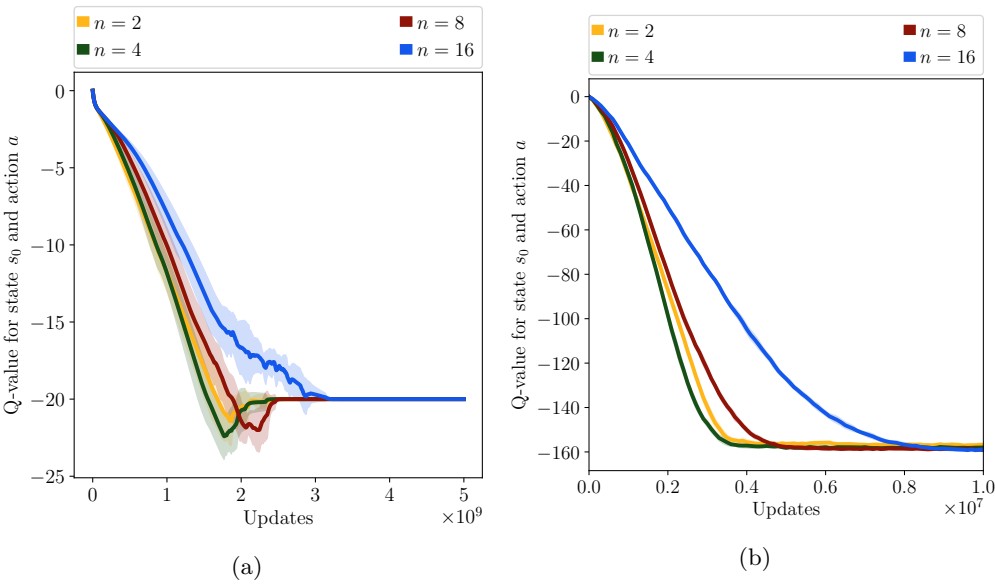

Figure B.1: (a) Performance of Composite Q-learning applied to the MDP in Figure 2a with different rollout lengths $n$. (b) Performance of Composite Q-learning applied to the MDP in Figure 4 with different rollout lengths $n$.

## C   Hyperparameter Setting

For all approaches, we use Gaussian noise with $\sigma = 0.15$ for exploration and the optimized learning rate of $10^{-3}$ for the full Q-function. Target update ($5 \cdot 10^{-3}$) and actor setting (two hidden layers with 400 and 300 neurons and ReLU activation) are set as in (Fujimoto et al., 2018). For Humanoid-v2, we use a slightly changed parameter setting with a learning rate of $10^{-4}$ for both actor and critic as suggested in (Dorka et al., 2020). For Composite Q-learning, we calculate the full Q-values, as well as Truncated and Shifted-Q values in one, combined, architecture for improved efficacy. Additionally, we found that it is beneficial to estimate the Truncated and Shifted Q-values in different layers, as depicted in Figure C.1. For all parameters in the layers prior to the full Q-output, we use the parameter setting as described above.

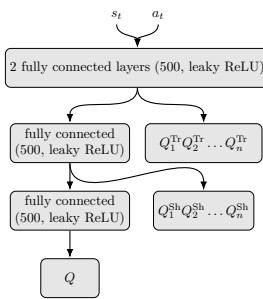

Figure C.1: Optimized architecture of the Composite-Q network used in our experiments.

Hyperparameters are optimized for all approaches, including the baselines TD3 and TD3($\Delta$), for the given subset of MuJoCo tasks via Random Search over 10 training runs with the configuration space shown in Table C.1. Learning rates for the corresponding output layers of the Truncated and Shifted Q-values are optimized individually. We use a learning rate of $6 \cdot 10^{-5}$ for the Truncated Q-functions and $5 \cdot 10^{-3}$ for the Shifted Q-functions. For the noisy experiments, as well as for the experiments with multiple gradient steps per collected sample, we set $n = 4$. For the Walker2d-v2 experiment on the true reward function, we set $n = 10$. In terms of regularization, we set $\beta_{\mathrm{Tr}} = 0.002$ and $\beta_{\mathrm{Sh}} = 0.001$. For Humanoid-v2, we set the optimized learning rates one magnitude lower. In the experiments with multiple gradient steps per collected sample, we execute five consecutive updates per collected sample and keep the learning rate of the Shifted Q-functions at $10^{-3}$. We further evaluate TD3 with the same maximum learning rate. We optimize the number of layers and the number of neurons per layer for the critic in Composite Q-learning (4 layers with 500 neurons and leaky ReLU activation) and the architecture for the critic in all other approaches (two layers with 500 neurons and leaky ReLU activation). For an overview of the architecture of the Composite-Q network see Figure C.1. For TD3($\Delta$), we use the $\gamma$-schedule as suggested by Romoff et al. (2019), i.e. $\gamma_1 = 0$ and $\gamma_{i>1} = \frac{\gamma_{i-1}+1}{2}$, with an upper limit of 0.99.

Table C.1: Configuration space of the hyperparameter optimization. For the full Q-function in all approaches, we used the optimized learning rate of $10^{-3}$ as in (Fujimoto et al., 2018). Hyperparameters denoted by $^*$ were optimized individually for all approaches.

| Hyperparameter | Values |
|---|---|
| number of layers* | $\{2, 3, 4\}$ |
| number of neurons* | $\{300, 400, 500\}$ |
| activation* | $\{\mathrm{ReLU}, \mathrm{leaky\ ReLU}\}$ |
| $\alpha_{\mathrm{Tr}}$ | $\{6 \cdot 10^{-6}, 10^{-5}, 6 \cdot 10^{-5}, 10^{-4}, 10^{-3}\}$ |
| $\alpha_{\mathrm{Sh}}$ | $\{10^{-3}, 2 \cdot 10^{-3}, 5 \cdot 10^{-3}, 10^{-2}\}$ |
| $\beta_{\mathrm{Tr}}$ | $\{10^{-3}, 2 \cdot 10^{-3}, 4 \cdot 10^{-3}\}$ |
| $\beta_{\mathrm{Sh}}$ | $\{5 \cdot 10^{-4}, 10^{-3}, 2 \cdot 10^{-3}\}$ |
| $n$ | $\{3, 4, 10, 15, 20\}$ |

## D  Significance Tests

We provide significance tests on the mean area under the learning curve (to account for learning speed and stability) between Composite TD3 and TD3, TD3 (HL) and TD3($\Delta$). Significance tests for the vanilla reward experiments can be found in Table D.1.

Table D.1: p-values for Welch's t-test on the mean area under the learning curve for 8 different runs of Composite TD3 and all other approaches in the vanilla reward experiments. Subparts (a) and (b) correspond to those in Figure 7.

|                  | TD3              | TD3 (HL)         | TD3($\Delta$)    |
| ---------------- | ---------------- | ---------------- | ---------------- |
| Walker2d-v2 (a)  | $8 \cdot 10^{-3}$ | $7 \cdot 10^{-4}$ | $6 \cdot 10^{-2}$ |
| Walker2d-v2 (b)  | $3 \cdot 10^{-4}$ | $2 \cdot 10^{-2}$ | $5 \cdot 10^{-1}$ |
| Hopper-v2 (b)    | $3 \cdot 10^{-5}$ | $4 \cdot 10^{-2}$ | $5 \cdot 10^{-1}$ |

Significance tests for the noisy reward experiments can be found in Table D.2.

Table D.2: p-values for Welch's t-test on the mean area under the learning curve for 8 different runs of Composite TD3 and all other approaches in the noisy reward experiments.

|              | TD3              | TD3 (HL)         | TD3($\Delta$)    |
| ------------ | ---------------- | ---------------- | ---------------- |
| Walker2d-v2  | $4 \cdot 10^{-5}$ | $2 \cdot 10^{-8}$ | $6 \cdot 10^{-8}$ |
| Hopper-v2    | $8 \cdot 10^{-3}$ | $8 \cdot 10^{-3}$ | $2 \cdot 10^{-6}$ |
| Humanoid-v2  | $3 \cdot 10^{-2}$ | $4 \cdot 10^{-4}$ | $5 \cdot 10^{-6}$ |

## E  Evaluation of Composite DQN on LunarLander-v2

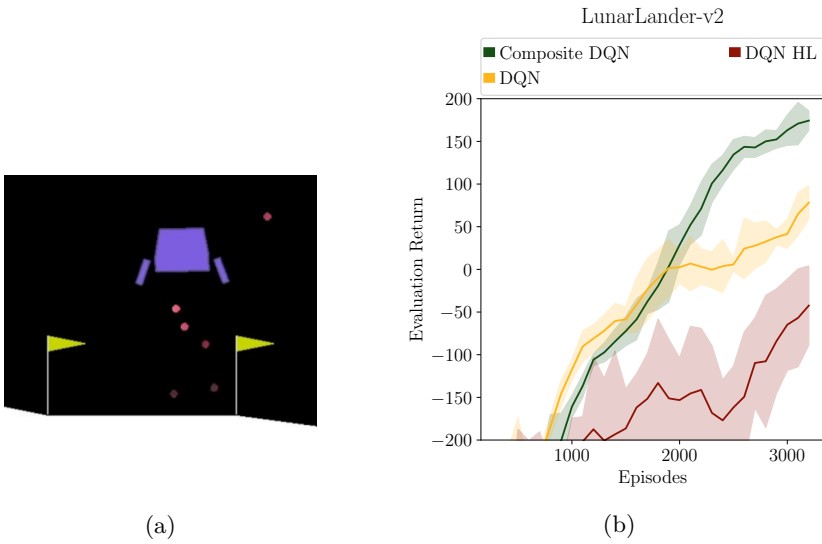

(a)                                                    (b)

Figure E.1: (a) Visualization of the LunarLander-v2 environment. (b) Mean performance and half a SD over 6 training runs for LunarLander-v2 with uniform noise on the reward function.

In addition to the continuous actor-critic experiments in Section 4, we evaluate Composite Q-learning within DQN on the LunarLander-v2 environment (shown in Figure E.1a). The immediate reward is again replaced by a uniform sample $u \sim \mathcal{U}[-1, 1]$ with 40% chance. We use a learning rate of $10^{-2}$ for the Shifted Q-functions and $10^{-5}$ for the Truncated and full Q-functions in Composite Q-learning. We compare to DQN

with a learning rate of $10^{-5}$ (DQN) and with a learning rate of $10^{-2}$ (DQN HL) in Figure E.1b. As can be seen, Composite Q-learning shows the same tendencies in the value-based method DQN as in the actor-critic method TD3.

## F  Comparison to Multi-step Baselines

Data-efficiency of *on-policy* Temporal-Difference methods can be enhanced by the use of $n$-step returns, where a Monte Carlo rollout of length $n$ is combined with a bootstrap of the value function. Reinforcement learning methods of this kind are called *multi-step*. Since our method takes up on a similar rationale, we lastly add a comparison to two multi-step methods.

To employ $n$-step returns in an *off-policy* setting, subtrajectories of the exploratory policy have to be stored. These stored multi-step returns, however, will differ from the true value of the target-policy. We evaluate this setting as *On-policy Multi-step TD3* in the following. To avoid the problem of the multi-step return being *off* the target-policy, a learned dynamics model can be used for imaginary rollouts, the so-called *Model-based Value Expansion* (MVE) (Feinberg et al., 2018). We hence employ Model-based Value Expansion within TD3 as additional baseline, subsequently called *MVE-TD3*.

We add Gaussian policy smoothing to the rollout of the model. In contrast to Feinberg et al. (2018), however, we do not assume to have knowledge about the reward function. Our model therefore approximates both, dynamics and reward. For MVE-TD3, we use a rollout length of three, as described in Feinberg et al. (2018) for the considered benchmarks. For *On-policy Multi-step TD3*, we use the same truncated horizons as for Composite TD3. Code for both multi-step baselines can be found in the supplementary[5].

Results for Walker2d-v2 with vanilla reward and a single update per collected sample, as well as results for Walker2d-v2 and Hopper-v2 with vanilla reward and five updates per sample are depicted in Figure F.1a and Figure F.1b, respectively. MVE-TD3 is highly sensitive with regard to accumulating errors in reward and state prediction, even when applying the TD-$k$ trick. The erroneous updates of *On-policy Multi-step TD3* are also causing harm to value-estimation, especially for the more stochastic Hopper-v2 environment. The results emphasize the importance of truly off-policy and model-free truncated value-estimation.

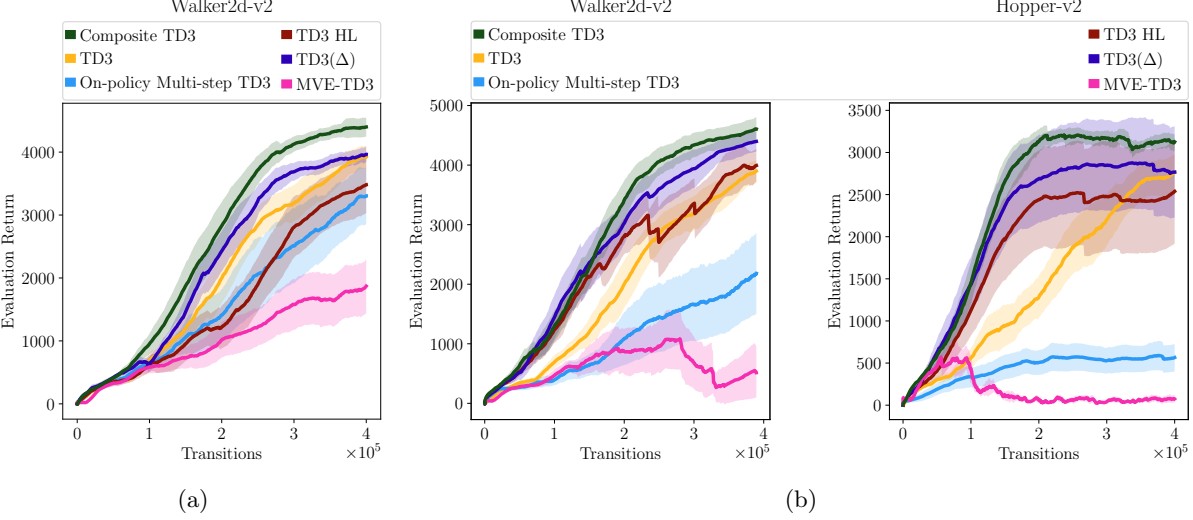

Figure F.1: (a) Mean performance and half a SD (standard deviation) over 8 training runs for the Walker2d-v2 environment with the default reward. (b) Mean performance and half a SD over 8 training runs of Composite Q-learning on the vanilla reward function and multiple update steps per collected sample for the Walker2d-v2 and Hopper-v2 environment.

---

[5]https://github.com/NrLabFreiburg/composite-q-learning

