# OpenReview forum: "Robust and Data-efficient Q-learning by Composite Value-estimation"
_TMLR — Accepted by TMLR_

### Review · Reviewer_uBm3 · 2022-05-11

**Summary Of Contributions:**

This paper introduces Composite Q-learning, a new reinforcement learning (RL) algorithm that combines two estimates of Q-values:
1) A "truncated" Q-function (esentially a discounted, episodic Q-value)
2) a "shifted" Q-function (standard Q-value, but starting from a state further in time).

The authors claim their method is able to deal with stochasticity in the environment better than prior methods, and is more robust to shifting learning rates, and argue for this by comparing against a strong existing baseline (TD3) on a few MuJoCo tasks.

The authors provide their code with their submission, which is always appreciated.

**Broader Impact Concerns:**

No real concerns here.

**Requested Changes:**

# Main suggestions
The suggestions marked as critical are those I feel are needed for me to recommend acceptance.
1. **Evaluate on value-based methods (critical)** Evaluate incorporating the algorithm on value-based methods such as DQN. (See "Weaknesses" above for details).
2. **Higher learning rates (critical)** Some of the analyses in the toy example (currently in the appendix) could be moved to the main paper, or at least discussed there. In particular, in a number of places in the main paper the authors claim their method is more robust to higher learning rates, but this is only really evaluated in the toy experiment (which is in the appendix). In addition, perhaps the authors could run CompTD3 with a higher learning rate for the MuJoCo tasks. In the figures in the main paper it appears they only tried a higher learning rate with TD3.
3. Equation (5) seems to imply that $Q^{Sh\prime}_{i-1}$s is learning the $\gamma^{i-1}$ multiplier, is this correct? If so, this should be made more explicit.
4. In section 4 the authors say "for Soft Actor-Critic the same results can be assumed to hold." No. Just because a method works well when added to one algorithm does _not_ mean it is guaranteed to hold on other algorithms. This would require actually running the experiments to verify this. I think running with value-based methods would be more critical, but running on SAC would strengthen the paper a lot more. At a minimum, I'd remove this sentence as there's no real justification for it.
5. **AUC values (critical)** The authors specify a percentage for AUC, but it's not clear what this percentage represents (e.g. Composite TD3 is reported to obtain 100% AUC). The authors say it is "mean normalized area under the learning curve", but it's not clear _what_ it's normalized against.

# Questions
1. In Figure 6 it seems Tr4 has more variance in the True Reward setting compared to the Noisy reward, which is counterintuitive. Why do we see this?
2. In Figure 6 why is there such a big difference in TD errors for Tr4 between true and noisy rewards?
3. In Figure C.5 why does Composite Q-learning have multiple learning rates in parentheses? Did they all have _identical_ learning curves? This seems improbable...

# Minor issues
Some minor issues that can (hopefully) help improve the paper:
1. Abstract line to: s/"in their application for robot control"/"in their application to robot control"/
2. In the introduction, first paragraph: s/"are still far away from being"/"are still far from being"
3. In the introduction, first paragraph: "their lack in learning stability caused by the stochastic nature of transitions in real-world settings". Stochasticity affects things, for sure, but it's not _the_ cause, just one of many.
4. In the introduction, first paragraph: s/"are then combined in a mutual recursive definition"/"are then combined in a recursive definition"/
5. Top of page 2: It's not clear what is meant by: "Since the values corresponding to larger discount values only represent the residual to the values of smaller discount."
6. Top of page 2: Better to not use acronyms ("w.r.t") and just write it out explicitly.
7. I would suggest moving Figure 3 to the top of the page, as there is currently a single (partial) sentence above it that is easy to miss.

**Strengths And Weaknesses:**

# Strengths
1. The paper is pretty well written, it was fairly easy to follow. There are a few things I would suggest improving, which I list below.
2. The experimental results are clearly presented, with confidence bounds, and evaluated over a number of different settings (in particular, adding stochasticity in the rewards).
3. The algorithm is clearly presented and discussed.

---

# Weaknesses

My main criticism of this paper is that, while the results seem compelling, it is not clear to me whether this is a general-purpose method that can help other type of algorithms and/or environments.

One thing that stood out to me is that the proposed method seems to be aimed at better _value estimation_, yet it is only evaluated on an actor-critic method where value estimation only affects the policy indirectly (e.g. the critic). It would be far more convincing if the authors could evaluate their method on value-based methods (such as DQN and offshoots), as here improper value estimation _directly_ affects the quality of the policy.

A few more specific concerns below.

## Theory

Although I appreciate the authors wanting to provide a theoretical backing for their proposal, the theoretical results presented in the appendix are quite weak. In fact, it seems they are just (re)deriving the definitions of truncated and shifted Q-values. In particular, Theorem A.3 is really just saying that the standard definition of Q-values (which is an infinite sum) can be split into two sums: a finite one, and what comes after. This is quite clear (it's just a property of sums) and doesn't really require "Theorems".

One related nit on this point: t the top of page 4, the authors say "Following Theorem A.1 in the appendix, we approximate $Q^*_n(s_t, a_t)$ off-policy via consecutive bootstrapping." Theorem A.1 is not about approximation, it's really just defining truncated Q-functions, so it's not clear what is meant here.

## Implementation details

There are some implementation details that are introduced but not completely justified. It's not clear how critical they are to the algorithm's performance, but I feel they merit further empirical analysis to properly understand.

1. It seems the authors are using a combination of multiple heads for estimating the truncated Q-functions (perhaps this is what the authors meant by "consecutive bootstrapping" above?). This is somewhat unorthodox and it's not clear how necessary this is. Without the multiple heads this becomes quite similar to n-step backups.
In page 5 the authors say "Composite Q-learning is equivalent to Q-learning if the learning rates for Truncated, Shifted and full Q-function are set to the same values." I don't think this is correct, precisely because of the use of multiple heads. It would be nice to evaluate the effect of the multiple heads on regular Q-learning (e.g. with the same learning rate values).
2. In Algorithm 1, the authors are adding $\xi$ to the action selection. Is this necessary to the algorithm? It seems orthogonal to the idea proposed. Perhaps this is something that is coming from TD3, in which case it would be good to highlight the differences between Algorithm 1 and TD3 (using colors can be effective for this).

---

> ### Author Response · Authors · 2022-05-24
> **Reply to Reviewer uBm3**
>
> We sincerely thank reviewer uBm3 for the valuable feedback and thoughtful review and will expand on each of the given points in turn below.
>
> 	"Although I appreciate the authors wanting to provide a theoretical backing for their proposal, the theoretical results presented in the appendix are quite weak. In fact, it seems they are just (re)deriving the definitions of truncated and shifted Q-values. In particular, Theorem A.3 is really just saying that the standard definition of Q-values (which is an infinite sum) can be split into two sums: a finite one, and what comes after. This is quite clear (it's just a property of sums) and doesn't really require "Theorems"."
>
> We agree that the term "Theorem" might be a bit strong and therefore weakened the term to "Proposition" in the current draft of the manuscript.
>
> 	"One related nit on this point: t the top of page 4, the authors say "Following Theorem A.1 in the appendix, we approximate off-policy via consecutive bootstrapping." Theorem A.1 is not about approximation, it's really just defining truncated Q-functions, so it's not clear what is meant here."
>
> The "approximation" was referring to the sampling based variant that comes after in the text. However, we agree that it was confusing and rephrased that in the paper.
>
> 	"It seems the authors are using a combination of multiple heads for estimating the truncated Q-functions (perhaps this is what the authors meant by "consecutive bootstrapping" above?). This is somewhat unorthodox and it's not clear how necessary this is."
>
> In the deep setting, we use multiple heads of the function approximator to estimate the different intermediate values of truncation and shifting. We used multiple heads in order to use less parameters, since it can be assumed that the different value-estimations (of different horizon) can benefit from shared features.
>
> 	"In Algorithm 1, the authors are adding $\xi$ to the action selection. Is this necessary to the algorithm? It seems orthogonal to the idea proposed. Perhaps this is something that is coming from TD3, in which case it would be good to highlight the differences between Algorithm 1 and TD3 (using colors can be effective for this)."
>
> The $\xi$ is the exploration noise coming from DDPG, since actor $\mu$ is a deterministic policy representing $\arg\max_a Q(s,a)$.
>
> 	"One thing that stood out to me is that the proposed method seems to be aimed at better value estimation, yet it is only evaluated on an actor-critic method where value estimation only affects the policy indirectly (e.g. the critic). It would be far more convincing if the authors could evaluate their method on value-based methods (such as DQN and offshoots), as here improper value estimation directly affects the quality of the policy. [...] Evaluate on value-based methods (critical)."
>
> Yes, we agree and we actually cover value-based methods in our tabular experiments (where we see similar tendencies as in the deep continuous setting). Since these results are very essential, we now moved the tabular evaluations to the main body of the paper (in accordance with yours and the other reviews). As a sidenote, we wanted to clarify that we build upon the deterministic actor-critic method DDPG in the continuous setting (or to be more precise on TD3) which is rather close to discrete deep Q-learning methods. We hope that this satisfactorily addresses the aforementioned weakness of the manuscript. Lastly, we added an evaluation of Composite DQN on the LunarLander-v2 environment with noisy rewards to the appendix (same tendencies as for Composite TD3 and tabular Composite Q-learning).

---

> > ### Author Response · Authors · 2022-05-24
> > **Reply to Reviewer uBm3**
> >
> > 	"Higher learning rates (critical) Some of the analyses in the toy example (currently in the appendix) could be moved to the main paper, or at least discussed there. In particular, in a number of places in the main paper the authors claim their method is more robust to higher learning rates, but this is only really evaluated in the toy experiment (which is in the appendix). In addition, perhaps the authors could run CompTD3 with a higher learning rate for the MuJoCo tasks. In the figures in the main paper it appears they only tried a higher learning rate with TD3."
> >
> > We actually applied a "higher" learning rate to Composite TD3 in the MuJoCo tasks. To clarify: We evaluated TD3 on multiple (low and high) learning rates, showing that TD3 cannot benefit from higher learning rates at some point. We then took the higher learning rate (which did not work for vanilla TD3), used it as the learning rate for the Shifted Q-functions within Composite TD3 and achieved very well performing policies. We hence argue that the extension of TD3 to Composite TD3 allows for higher learning rates compared to the vanilla version, at least for some of the individual parts of value-estimation. We added a clarifying comment and furthermore moved the tabular experiments to the main paper.
> >
> > 	"Equation (5) seems to imply that $Q^\text{Sh}'_{i-1}$ is learning the $\gamma^{i-1}$ multiplier, is this correct? If so, this should be made more explicit."
> >
> > Yes, this is correct and we added a comment in the paper.
> >
> > 	"In section 4 the authors say "for Soft Actor-Critic the same results can be assumed to hold." No. Just because a method works well when added to one algorithm does not mean it is guaranteed to hold on other algorithms. This would require actually running the experiments to verify this. I think running with value-based methods would be more critical, but running on SAC would strengthen the paper a lot more. At a minimum, I'd remove this sentence as there's no real justification for it."
> >
> > While we were referring to the work of Ball and Roberts (2021), we agree that this is an unproven statement and therefore removed it.
> >
> > 	"AUC values (critical) The authors specify a percentage for AUC, but it's not clear what this percentage represents (e.g. Composite TD3 is reported to obtain 100% AUC). The authors say it is "mean normalized area under the learning curve", but it's not clear what it's normalized against."
> >
> > We normalized by the mean AUC of Composite TD3 to set the mean performance of Composite TD3 as the origin for comparison. We hence only rescaled the resulting AUCs and did not put Composite TD3 on any advantage. We added a clarifying comment in the paper.
> >
> > 	"In Figure 6 it seems Tr4 has more variance in the True Reward setting compared to the Noisy reward, which is counterintuitive. Why do we see this?"
> >
> > The stronger fluctuation of Tr4 is possibly the result of premature overfitting of the Truncated Q-functions to the non-noisy rewards in the very beginning. The larger variance of Sh4 could be explained by the generally broader range of returns the policy already collects at an earlier stage of training in the true reward setting.
> >
> > 	"In Figure 6 why is there such a big difference in TD errors for Tr4 between true and noisy rewards?"
> >
> > The noise underlying the immediate reward possibly induces the different intermediate steps of truncated value-estimation to disagree which can lead to higher errors. Since the shifted value-functions skip the first n noisy rewards, they are less prone to this issue.
> >
> > 	"In Figure C.5 why does Composite Q-learning have multiple learning rates in parentheses? Did they all have identical learning curves? This seems improbable..."
> >
> > There is only one experiment and hence only one learning curve shown in Figure C.5. Composite Q-learning allows for an individual setting of learning rates for the full Q-value, the truncated Q-value and the shifted Q-value. The three learning rates correspond to the learning rates of the different components (full, shifted, truncated). We compare against Q-learning evaluated with all of the three learning rates. We clarified this in the caption.
> >
> > 	"Minor issues"
> >
> > Lastly, we incorporated all suggestions from "Minor issues".

---

### Review · Reviewer_qbp7 · 2022-05-15

**Summary Of Contributions:**

This paper proposes Composite Q-learning, a conceptually clean new variant of Q-learning that combines the estimation of two kinds of Q-functions: (1) The "Truncated" Q function, and (2) the "Shifted" Q function. The Truncated Q function is an n-step roll out return, while the Shifted Q function is an estimate of the return _after_ this n step rollout. Composite Q-learning, abstractly, is a generalization of Q-learning in which different learning rates (and hence optimization schedules) can be applied to each of these constituent Q-functions. The paper suggests exposing this extra degree of control over the optimization process, variance can be controlled to a greater degree, thus allowing for the use of higher learning rates (particularly in environments with stochastic rewards, or rewards that depend on a stochastic future state). These insights are combined into a few algorithmic variations of the central idea, including TD3($\Delta$), that extends TD$(\Delta)$ to off-policy learning. Thorough experiments are conducted in simulated control tasks such as Walker and Hopper in two varieties: the first uses a clean reward signal, while the second uses a noisy reward signal (following work by Romoff et al. 2019). Data are examined and presented carefully, with the primary evidence supporting the claim that Composite Q-learning can be particularly data-efficient under reward noise.

**Requested Changes:**

**Reactions**

I was left with one question. I do not believe the authors need to address this question in the paper by any means, but I thought it might be a useful question to think about:
- Does further temporal decomposition help? That is, Composite Q-learning decomposes Q into the first n steps, then the future value. This extra granularity allows for greater control over the learning process and parameter schedules. But, Equation 2 is suggestive that there might be _arbitrarily_ many such distinctions; is it valuable to continue going beyond a single split? That is, might we consider a generalization in which we maintain 3 Q estimates: $Q_{[1:n]}, Q_{[n:n+k]}, Q_{[n+k:\infty]}$?  We might generalize this further to arbitrarily many such distinctions. I was curious if the authors had thought about any potential advantages from such an approach.

I have three primary recommendations for improving the paper:
- Recommendation 1: I believe more discussion of both Figure 4 and Figure 6 could strengthen the work, as it seems there could be many insights we might draw here. Can you expand on what you take to be the important findings from these data? The discussion of Figure 4 in particular is quite brief at the moment.
- Recommendation 2: I was slightly surprised to see Theorems in the appendix my first read through. Are any of these results central enough to include in the main text?
- Recommendation 3: I found many of the experiments in the appendix to be quite insightful. If the authors find space, I believe they could have a more natural home in the main text of the paper (perhaps just C.1, if space is really tight).


**Writing Suggestions.** I have a few low-level writing suggestions:

1. Introduction:
- "environments of higher complexity": Higher complexity than what? Perhaps "of high complexity" would be more suitable here.

2. Background
- The $\mapsto$ symbol should be changed to $\rightarrow$ in the introduction of the MDP components. "$\mathcal{X} \rightarrow \mathcal{Y}$" is used to express that a function takes as input elements $x \in \mathcal{Y}$ and outputs elements $y \in \mathcal{Y}$, whereas "$\mapsto$" actually describes the map: $f : x \mapsto x^2$, for instance, shows that $f(x) = x^2$.
- I find the notation $\mathcal{R}$ for the expected return to be non-standard compared with calling this quantity the "value" function, expressed with either $v$ or $V$.

3. Combining...

- The superscript and subscript notation became hard to parse quite quickly, starting around Equation 4, and especially Equation 5 on. I very quickly lost track of the precise meaning of what $\theta^{Q'}$, $y_j^Q$, $Q^{Tr}$, and found myself regularly going back to double check their definitions. In particular, I found parsing Algorithm 1 to be extremely time consuming. I have two recommendations: (1) Add vertical spacing between the lines of the pseudocode to prevent the superscripts/subscripts from overlapping (under the "calculate targets" section), and (2) Consider alternate forms of notation apart from superscripts, such as hats/overlines or changes in letter/font.

4. Experiments:
- Just to be really precise, it would be useful to define the acronym "SD" when it is first used.
- A personal preference, but I find the use of bolding in tables when the confidence intervals overlap to a large degree to be slightly misleading, and would encourage removing the bold (or just explaining its use in the caption).

Bibliography:
- A few titles should be capitalized: "Averaged-dqn" --> "Averaged-DQN", "Openai gym" --> "OpenAI gym", and "...and go" --> "...and Go".
- Just a personal preference, but the citation format for conferences is slightly inconsistent throughout. For instance, "in 4th International Conference on Learning Representations, ICLR, 2016."

**Strengths And Weaknesses:**

**Strengths.** There are many things to praise about this paper. Succinctly:
1. _Algorithmic Simplicity_: The main idea giving rise to Composite Q-learning is intuitive, well-explained, and ultimately simple enough that I believe the community can iterate and experiment with it quite readily.
2. _Clarity_: The paper and appendix are well presented, taking its time to expose the key insights that constitute Composite Q-learning. I have some small suggestions for improving clarity which I discuss below.
3. _Experimentation_: The experiments, by my reading, are very sensibly-motivated and well-conducted. Results tell a clear story and the claims made in the paper are supported by the empirical evidence. In particular I found Figure 4 and Figure 6 to be very interesting plots.

To summarize: I believe the contributions in the paper are both of interest to members of the ML community, and the main claims are well supported.

**Weaknesses.**
1. _Notation (Pseudocode and Superscripts)_: As I detail below in the writing suggestions, I found some of the notation to be overly complicated at the detriment of clarity in some passages. In particular, the use of subscripts and superscripts in some of the Q notation, and especially Algorithm 1, can be much improved.
2. _Discussion_: I believe some of the experimental section could include further discussion drawing out take aways and insights from the results (specifically Figure 4 and 6---further comments given below).

---

> ### Author Response · Authors · 2022-05-24
> **Reply to Reviewer qbp7**
>
> We highly appreciate the feedback from reviewer qbp7 and want to thank for the insightful comments and suggestions. In the following, we will respond to each of the aforementioned points.
>
> 	"Notation (Pseudocode and Superscripts): As I detail below in the writing suggestions, I found some of the notation to be overly complicated at the detriment of clarity in some passages. In particular, the use of subscripts and superscripts in some of the Q notation, and especially Algorithm 1, can be much improved."
>
> We totally agree that notation became quite involved and may appear unreasonably complicated, however we actually tried different notations until we arrived at the one shown in the manuscript and it seemed to be the best compromise. However, we added vertical spacing to improve upon readability of the algorithms.
>
> 	"Does further temporal decomposition help? That is, Composite Q-learning decomposes Q into the first n steps, then the future value. This extra granularity allows for greater control over the learning process and parameter schedules."
>
> This is a great suggestion and we agree that this could be a promising avenue to provide even more flexibility. This, however, then comes at the cost of more hyperparameters, which means that more sophisticated HPO and meta-learning methods could become of high importance. In principle, further decomposition would be possible by shifting a truncated value-function in time. We added a comment to the paper.
>
> 	"Recommendation 1: I believe more discussion of both Figure 4 and Figure 6 could strengthen the work, as it seems there could be many insights we might draw here. Can you expand on what you take to be the important findings from these data? The discussion of Figure 4 in particular is quite brief at the moment."
>
> We totally agree and added a broader discussion of Figure 4 and 6 to the paper.
>
> 	"Recommendation 2: I was slightly surprised to see Theorems in the appendix my first read through. Are any of these results central enough to include in the main text?"
>
> While we agree that a proof of correctness of the consecutive bootstrapping scheme is of high importance, we initially thought that it is clear enough to be put in the appendix. However, after consideration of all the reviews, we included the theorems now in the paper, left the proofs in the appendix and weakened the term to "proposition".
>
> 	"Recommendation 3: I found many of the experiments in the appendix to be quite insightful. If the authors find space, I believe they could have a more natural home in the main text of the paper (perhaps just C.1, if space is really tight)."
>
> Yes, we totally agree and added the evaluation of tabular Composite Q-learning to the main paper.
>
> 	"Low-level writing suggestions"
>
> We included all of the writing suggestions.

---

> > ### Comment · Reviewer_qbp7 · 2022-06-12
> > **Response to Authors**
> >
> > I thank the authors for their thorough replies to each of the reviews. I believe the paper has been strengthened due to the exchange, and I maintain a positive endorsement of the work.

---

### Review · Reviewer_ZC7n · 2022-05-15

**Summary Of Contributions:**

This paper proposes a composite Q learning algorithm where the temporal difference (TD) update is decomposed into multiple truncated Q functions, the returns of which are based on a fixed temporal horizon. The authors argue that fixing the temporal horizon, and using a decomposition technique to then combine the multiple Q functions together, is a useful approach to handle robustness, in presence of noise in reward signals or if the environment is stochastic. As a bi-product, different hyperparameters can be used since the composition is based on two different Q estimations(based on truncation and shifting of Q values).

The authors try to demonstrate these claims through a series of experiments, both for continuous control and toy discrete action domains; while the theoretical results argue that even under this compositionality, the full return as in a standard Q learning algorithm is still maintained (thereby, existing theoretical convergence results, or divergence counterexamples would still hold). The main benefit of the compositionality is the use-case in presence of noise reward signals (which demands study in the field to enhance practicality of existing RL algorithms) or when environment is stochastic (where robustness is of importance). The resulting algorithm is mainly integrated into a well known, state of the art TD3 algorithm and experiments demonstrate that this plug-in addition of composite Q values, in the critic estimation of TD3 can be a useful technique.

From an algorithm and methodology perspective, the paper does propose a new approach which can be useful, although it comes with its own drawbacks of being able to compute, possibly with different function approximators too, for the different Q functions. The ability to decompose n-step returns into multiple Q functions has been studied in the past, but the paper proposes a different technique of doing this. The practical usefulness of it, and whether it sufficiently demonstrates the claims for which this technique is useful, is perhaps questionable and would require further study. Please see comments below.




**Requested Changes:**

	1. The experiment comparisons with a 1-step version seems concerning to me; it is not clear what to expect if we could compare with a n-step return baseline too, since composite TD3 clearly uses n-step returns.

Overall, my major comment would be that the proposed claim of being able to handle stochasticity is perhaps not well established. It is quite clearly shown in the toy tasks, but not on some of the main tasks of the paper. I am ok with if the toy-ish tasks are included as main draft in the paper - because clearly it does demonstrate the usefulness of the approach there.

I would suggest a careful re-write of the paper in terms of how the proposed claims are demonstrated would indeed be helpful for the wider audience.  Otherwise, the paper is well written and easy to follow.


**Strengths And Weaknesses:**

Strengths :

	1. The paper is very clearly written and the methodology, including the figure explaining the approach was very helpful. It has a very clear explanation of the idea, and is brutally honest about what is novel and not. I like the fact that authors even include an off-policy baseline TD3 (Delta) based on modification of their approach and an existing work.

	2. The theoretical claims seem justified; although some analysis of the bias variance trade off would probably be useful. The theoretical results seem to justify the use of achieving lower variance returns too, which is helpful but perhaps require more clarity. The n-step returns that composite Q depends on clearly has a bias variance trade-off depending on the choice of n.

	3. Experiment results are well demonstrated and clearly shows significance in standard control tasks, and more so when additional noise is introduced to rewards. The benefits seem obvious, and I will not complain much about the use of baselines, except the fact that future iterations of the paper probably needs some fair comparisons with baselines that also uses n-step returns (I am ok with if that is not demonstrated in a control task, but in some other domain - see comments below).

	4. The clarity in the paper, and clearly showing what the algorithm does, and the illustration of the approach is appreciable.

	5. Algorithmically I think the paper has novelty, and if demonstrated well in some other stochastic environments, then this can be a useful approach. Although it is easy to complain that this paper clearly builds from existing works of Romoff et al., - I think to make this kind of approach work in practice is clearly a good contribution of the paper. I am just not convinced whether the claims are enough experimentally supported - maybe the approach has other benefits, and as such a careful re-writing of the claims would perhaps be helpful.



Weaknesses:

	1. I am worried about the comparisons of baselines somewhat - mainly due to the fact that the baselines don't all seem to use n-step returns, and rather uses 1-step return for the off-policy counterpart? Do the authors have a good sense of how this might affect results? I understand that the proposed method needs to be evaluated for n-step returns, while the baseline being off-policy needs to be done for 1-step return. Why not include other experiment comparions which uses some version of V-traces or other TD(lambda) based algorithms that are applicable for control? There used to be an ACER algorithm for control too, which uses a n-step return version for the critic in control task - which might be helpful.

	2. The proposed claim of the paper is to improve robustness in presence of stochastic environments or sparse reward signals. I am not fully convinced that the proposed claims are justified through the series of experiments? The control environments are all deterministic; simply adding noise to reward signals does not seem like a very well justified approach?


3. I am not sure whether the limitation of applying composite Q learning has to be based on control experiments only. Clearly the method can be general and can be applied widely in any existing off-line setup. It might be helpful to think about a different domain where the claim of "robustness to stochasticity" can be better demonstrated?

---

> ### Author Response · Authors · 2022-05-24
> **Reply to Reviewer ZC7n**
>
> First and foremost, we want to express our gratitude to reviewer ZC7n for the in-depth review and valuable suggestions. In this comment, we want to provide a reply to each of the given points.
>
> 	"1. I am worried about the comparisons of baselines somewhat - mainly due to the fact that the baselines don't all seem to use n-step returns, and rather uses 1-step return for the off-policy counterpart? Do the authors have a good sense of how this might affect results? I understand that the proposed method needs to be evaluated for n-step returns, while the baseline being off-policy needs to be done for 1-step return. Why not include other experiment comparions which uses some version of V-traces or other TD(lambda) based algorithms that are applicable for control? There used to be an ACER algorithm for control too, which uses a n-step return version for the critic in control task - which might be helpful."
>
> We actually do not use n-step returns, but rather learn truncated and shifted Q-values from single-step transitions alone. We therefore decided to compare against other single-step approaches, since we believe this to be a fairer and clearer comparison. Otherwise, the respective data sets the algorithms learn from would be very different which would even more harm comparability. We added a clarifying comment to the paper.
>
> 	"The proposed claim of the paper is to improve robustness in presence of stochastic environments or sparse reward signals. I am not fully convinced that the proposed claims are justified through the series of experiments? The control environments are all deterministic; simply adding noise to reward signals does not seem like a very well justified approach?"
>
> We agree that the wording was partially misleading and therefore removed the claim that Composite Q-learning increases robustness against stochastic environments. While stochastic environments can be one cause of noisy immediate rewards, only the latter is backed by the experiments. In addition, we actually do not claim the approach to add benefit to sparse reward settings (as this mainly addresses exploration not tackled by this paper).
>
> 	"I am not sure whether the limitation of applying composite Q learning has to be based on control experiments only. Clearly the method can be general and can be applied widely in any existing off-line setup. It might be helpful to think about a different domain where the claim of "robustness to stochasticity" can be better demonstrated?"
>
> We agree that there are certainly other use cases apart from robotics and added a comment in the paper. However, we wanted to establish the approach on the basis of widely used, well-known and accepted benchmarks.
>
> 	"Overall, my major comment would be that the proposed claim of being able to handle stochasticity is perhaps not well established. It is quite clearly shown in the toy tasks, but not on some of the main tasks of the paper. I am ok with if the toy-ish tasks are included as main draft in the paper - because clearly it does demonstrate the usefulness of the approach there."
>
> We agree that the tabular experiments are of very high importance and therefore moved the respective sections to the main paper.

---

### Author Response · Authors · 2022-05-24
**Changes Since Last Submission**

First and foremost, we again want to thank the reviewers for their efforts in providing very valuable, thoughtful and detailed reviews. We furthermore want to thank TMLR for the exceptionally enjoyable review process.

We incorporated the feedback from all three reviewers. Most notably, we did the following modifications to the paper:

	* We moved the tabular derivation and evaluation of Composite Q-learning to the main paper.
	* We clarified the claims.
	* We improved upon clarity of the derivation.
	* We added an evaluation of Composite Q-learning within DQN to the appendix.
	* We added a broader discussion of Figures 4 and 6 (now Figures 8 and 10 in the current manuscript).

More detailed replies can be found in the comments to the respective reviews. Please let us know if you demand for further modifications, thank you for your effort.

---

> ### Comment · Reviewer_uBm3 · 2022-06-01
> **Response to latest suggestion**
>
> Thank you for the edits, they definitely helped improve the clarity of the paper's contributions, and I believe will be easier for future research to build on.
>
> Some minor suggestions based on the latest changes:
> 1. Above equation (15): "More detailed, ..." -> "More specifically, ..."
> 2. Above Figure 6: "... Q-learning as origin ..." -> "... Q-learning as the origin ..."
> 3. Bottom of page13: "is given exemplary for the Walker2d-v2" -> "is exemplified for the Walker2d-v2"
> 4. Bottom of page13: "The same holds for a too long temporal horizon." -> "The same holds for a temporal horizon that is too long."

---

> > ### Author Response · Authors · 2022-06-02
> > **Response to minor suggestions**
> >
> > We sincerely thank Reviewer uBm3 for the thoughtful comments. We included all minor suggestions in the current revision of the manuscript.

---

### Decision · Action_Editors · 2022-06-25

**Recommendation:** Accept as is

**Comment:**

The paper proposes two TD formulations: truncated Q-functions and shifted Q-functions, which together allow composite Q-learning for jointly learning Q-values across different temporal horizons from only single-step transitions. The method shows improved sample efficiency and more robustness to stochastic rewards, on simulated toy control environments.

Pros:
- All reviewers approved the clarity and simplicity of the presented methods. This could be a good educational paper for RL researchers looking into novel formulations of TD learning, especially bridging single-step and n-step approaches.
- The authors provided sufficient revisions, particularly re-organizing sections of the paper for clarity, and toning down claims originally made about stochastic environments. This helped better align claims and results included in the paper.

Cons:
- I agree with ZC7n that this paper as it stands does not have significant empirical impact. Not likely that many (deep) RL practitioners will switch to use this algorithm.
- Some n-step baseline is highly recommended to be added, e.g. TD3 + n-step without importance correction.
- We did not reach consensus (two accepts, one lean to reject)

The final recommendation of acceptance is based on the promising educational value of this paper, for its clear and simple derivations and analyses; however, we also note that empirical value of this paper is limited due to lack of comparisons (e.g. with n-step baselines) and evaluations on harder/more realistic domains. We encourage the authors to further improve the work, such that it can benefit wider audience of TMLR.